# Sequencing DNA with nanopores: Troubles and biases

**Clara Delahaye** [ID]ᵒ, **Jacques Nicolas**ᵒ*

Inria, CNRS, IRISA, Univ Rennes, Rennes, France

ᵒ These authors contributed equally to this work.
* jacques.nicolas@inria.fr

**Data Availability Statement:** All relevant data are within the manuscript and its Supporting information files.

**Funding:** The author(s) received no specific funding for this work.

## Abstract

Oxford Nanopore Technologies' (ONT) long read sequencers offer access to longer DNA fragments than previous sequencer generations, at the cost of a higher error rate. While many papers have studied read correction methods, few have addressed the detailed characterization of observed errors, a task complicated by frequent changes in chemistry and software in ONT technology. The MinION sequencer is now more stable and this paper proposes an up-to-date view of its error landscape, using the most mature flowcell and basecaller. We studied Nanopore sequencing error biases on both bacterial and human DNA reads. We found that, although Nanopore sequencing is expected not to suffer from GC bias, it is a crucial parameter with respect to errors. In particular, low-GC reads have fewer errors than high-GC reads (about 6% and 8% respectively). The error profile for homopolymeric regions or regions with short repeats, the source of about half of all sequencing errors, also depends on the GC rate and mainly shows deletions, although there are some reads with long insertions. Another interesting finding is that the quality measure, although over-estimated, offers valuable information to predict the error rate as well as the abundance of reads. We supplemented this study with an analysis of a rapeseed RNA read set and shown a higher level of errors with a higher level of deletion in these data. Finally, we have implemented an open source pipeline for long-term monitoring of the error profile, which enables users to easily compute various analysis presented in this work, including for future developments of the sequencing device. Overall, we hope this work will provide a basis for the design of better error-correction methods.

## Introduction

Nanopore sequencing is based on measuring changes in the electrical signal generated from DNA or RNA molecules passing through nano-scaled pores. This third-generation technology is developed and marketed by Oxford Nanopore Technologies (ONT), that uses a small portable sequencing device called MinION [1]. It offers many interesting features, including long read sequencing (the mean read length often exceeds 10 kb, and maximal read length now reaches up to 880 kb [2]), a real-time analysis and a low initial investment.

**Competing interests:** The authors have declared
that no competing interests exist.

However, although this technology has improved over recent years, it still exhibits a relatively high error rate on raw sequences compared to standard Next-Generation Sequencing (NGS) devices such as Illumina. In the Phase 1 early access program from Nanopore, a study from the MinION Analysis and Reference Consortium [3] showed that the 2D pass reads had a total error of 10.5%, including about 3% for mismatch and insertion and slightly more for deletion. The software in charge of the translation from signal to nucleic sequences, the basecaller, has proven to be crucial over the years for the accuracy of the resulting raw read sequences [4]. With the current technology and the most mature basecaller, and with great variations depending on organisms and read quality, the current mean global error rate on raw reads seems to be around 6% for quality scores at least equal to 10 (the basecaller filters reads whose quality scores are below a certain threshold).

Many papers have studied ways to reduce the error rate of long read sequencing (either ONT or non-CCS (Circular Consensus Sequencing) Pacbio) by computing consensus sequences over subsets of reads. In fact, there is even a tool to evaluate error correction methods [5]. The standard approach is hybrid correction, making use of both long read and short read data to reduce errors [6–9]. It is very demanding since it requires two sources of sequence data. It is also possible to rely solely on the information contained in the long reads (self-correction) as long as there is sufficient sequencing depth (30–50X in practice). These tools are generally associated with assemblers (*e.g.* Sparc for Canu [10]) and based on a mapping phase to detect overlapping reads.

For tasks involving a fine investigation of natural variations occurring among copies (*e.g.* genotyping or haplotyping), it is however important to limit the construction of consensus sequences and thus obtain better knowledge of the types of error occurring due to technological limits. For instance, Nanopore sequencers tend to struggle to sequence low complexity regions accurately (minor variation in the electrical signal of the pore when the base does not change). Since the DNA translocation speed is not constant, this results in difficulties determining the exact length of homopolymers. In contrast to error-correction methods, relatively few papers have proposed to delimit the landscape of errors made by the MinION and other sequencers of the Nanopore series. This knowledge is important but not publicly available, as Nanopore does not publish its benchmarks and software content. The task is made difficult by the rapid changeover of equipment, chemistry and programs [11]. Legget *et al.* have proposed an open-source software, NanoOK, to compare sets of references versus reads and produce an alignment-based analysis of errors and quality [12]. They look for indel and substitution errors, and analyze k-mers that are either error-free or over-/under-represented before error positions. One of the best recent study on errors [4], by R. Wick *et al.*, considers various measures on sequences issued from different basecallers. Since the Nanopore technology becomes more mature and stable, it seems useful to get a more accurate picture of the differences between known reference genomes and sequences extracted from MinION data, using the state-of-the-art basecaller. In this paper, we have worked on data produced by the primary nanopore used, R9.4.1. The new nanopore chemistry R10.3 is designed to improve homopolymer recognition, and thus the consensus accuracy. However, according to Nanopore, this chemistry still has some limitations compared to R9.4.1: it requires higher input (25–75fmol for R10.3, 5–50fmol for R9.4.1) and has lower output while the raw accuracy is similar to R9.4.1. Moreover, the R10.3 is only compatible with one sequencing kit, and does not handle methylation detection. We started this work with the basecaller Guppy version 3.3.3, the most widely used so far. Recent developments since then mostly concern extensions for improved detection of barcodes, faster runtime, and different environments (*e.g.* different GPU cards or PromethION instead of MinION). We also considered version 4.2.2, the most recent version at the time of writing, which improves accuracy by 1% compared to version 3.3.3. We will present the results

for this latest release while pointing out the differences with the other one whenever necessary (failing which, the reader can safely assume that the behaviors are the same).

We worked on a benchmark of 12 bacterial and two human datasets, sequenced with Nanopore MinION, in order to provide an overview of MinION sequencing errors on both prokaryotes and eukaryotes. We also evaluated several sequencing biases, in particular concerning the GC content of sequences, and repeated regions (homopolymers, heteropolymers, and trinucleotides).

We focused our analysis on DNA sequencing, however we also studied RNA sequencing data from rapeseed to provide an overview of the associated error rates.

## Materials and methods

### DNA and RNA read datasets

We used sets of DNA reads from various bacterial species and two human samples, which had all been sequenced with Nanopore MinION without any prior PCR amplification. This prevents errors due solely to the amplification phase from being considered. Part of the bacterial datasets came from our own sequencing experiments, with the others originating from Wick *et al.* 2019 [4], and human sequencing data come from Shafin *et al.* 2019 [13]. Details about all datasets are presented in Table 1. The associated reference bacterial genomes have various lengths and GC contents. In the remainder of this paper, bacterial species with less (resp. more) than 50% of GC content will be referred as "low (resp. high) GC" species.

**Experimental dataset**. We used sequencing data of *Streptococcus thermophilus* strains CNRZ1066 and LMD-9 (see Data Availability section). We used reference genomes that were available on NCBI: NC_006449.1 for *Streptococcus thermophilus* CNRZ1066, NC_008532.1 for *Streptococcus thermophilus* LMD-9.

**Wick *et al.* dataset**. These bacterial data gather sequencing of seven species: *Acinetobacter pitii*, *Haemophilus haemolyticus*, *Klebsiella pneumoniae* (four strains: INF032, INF042,

**Table 1. List of studied datasets.**

| Species (strain, if known) | Size (non-N bases) | GC % | Flowcell version | Mean read length (bp) | Sequencing depth | Source |
|---|---|---|---|---|---|---|
| *A. pittii* | 3,814,719 | 38.78 | R9.4.1 | 25,487 | 25X | Wick *et al.* 2019 |
| *H. haemolyticus* | 2,042,591 | **38.45** | R9.4.1 | 9,557 | 14X | |
| *K. pneumoniae* (INF032) | 5,111,537 | 57.63 | R9.4 | 35,886 | 69X | |
| *K. pneumoniae* (INF042) | 5,337,491 | 57.41 | R9.4 | 47,390 | 63X | |
| *K. pneumoniae* (KSB2 1B) | 5,228,889 | 57.59 | R9.4 | 23,620 | 46X | |
| *K. pneumoniae* (NUH29) | 5,134,281 | 57.61 | R9.4 | 15,851 | 33X | |
| *S. aureus* | 2,902,076 | 32.85 | R9.4.1 | 21,591 | 77X | |
| *S. maltophilia* | 4,802,733 | **66.28** | R9.4.1 | 30,694 | 69X | |
| *S. marcescens* | 5,517,578 | 59.13 | R9.4.1 | 7,729 | 19X | |
| *S. sonnei* | 4,829,160 | 51.03 | R9.4 | 20,313 | 65X | |
| Human HG002 | **3,110,720,511** | 41.04 | R9.4 | 19,290 | 29X | Shafin *et al.* 2019 |
| Human HG00733 | **3,110,720,511** | 41.04 | R9.4 | 14,731 | 28X | |
| *S. thermophilus* (CNRZ1066) | **1,796,226** | 39.08 | R9.4 | 6,049 | 330X | Own data |
| *S. thermophilus* (LMD-9) | 1,856,368 | 39.08 | R9.4 | 7,332 | 258X | |

Size and GC content are computed on the reference genome, highest and lowest values are in bold. Mean read length is computed for aligned reads basecalled with Guppy 4.2.2 with the HAC mode, and is rounded up to the nearest integer. It is slightly lower (a few hundred bases) than for Guppy 3.3.3. Sequencing depth is computed as the sum of all aligned sequenced bases divided by the size of the reference genome. It is generally greater than for Guppy 3.3.3.

KSB2 1B and NUH29), *Staphylococcus aureus*, *Stenotrophomonas maltophilia*, *Serratia mar-
cescens* and *Shigella sonnei*. All of these datasets were extracted and sequenced as described
in [14]. For each species, a reference genome generated from Illumina reads was also pro-
vided (see Data Availability section for fast5 files and reference genomes).

**Shafin *et al*. dataset**. We used two human sequencing data: HG002 and HG00733 (see Data
Availability section). Details about DNA extraction and sequencing are provided in [13].
We used the assembly GRCh38.p13 (GenBank assembly accession: GCA_000001405.28) as
reference genome.

In addition to DNA sequencing analysis, we conducted a quick analysis of sequencing
errors for RNA data. We used sequencing data of direct RNA sequencing *Brassica napus*
Darmor-bzh [15]. We asked the authors for fast5 files, which we then basecalled with Guppy
4.2.2. As reference, we used the corrected version of these reads, also kindly provided by the
authors. The mean read length is 691 bp and GC content is 43.82%. We focused our analysis
on DNA sequencing data, thus in the remaining of this paper, we will state explicitly when we
refer to RNA sequencing data, failing which the reader can safely consider that we refer to
DNA sequencing datasets.

### Basecalling and alignment

Each dataset was basecalled using Guppy on the fast5 files containing raw signals emitted by
nanopores. We used Guppy versions 3.3.3 (released on December $10^{th}$ 2019) and 4.2.2
(released on September $28^{th}$ 2020), the latter having a 1% improved modal accuracy compared
to the former. Guppy currently provides two main modes: a HAC (High ACcuracy) mode and
a FAST mode, which is a simplified version in which the computation time and accuracy are
reduced. Note that at present, there does not exist any public repository of these critical soft-
ware packages and their specifications: all this material can only be accessed through the
Nanopore community website, which requires an account (https://community.nanoporetech.
com/attachments/3640/download). Although most of the results presented relate to the HAC
mode, we used both HAC and FAST, as well as a specific basecalling mode designed to
retrieve methylation information (6mA dam and 5mC dcm methylation for bacterial data,
5mC CpG methylation for human data). We used the following configuration files (provided
with Guppy basecaller) for the different basecalling modes: `dna_r9.4.1_450bps_hac.
cfg` for HAC mode, `dna_r9.4.1_450bps_fast.cfg` for FAST mode, and `dna_
r9.4.1_450bps_modbases_dam-dcm-cpg_hac.cfg` for methylation aware mode.

ONT recommends to keep only reads with a quality score above 7. We chose to be more
stringent (see section *Correlation between error rate and amount of reads* in Results) and to set
this threshold to 10 (`--qscore_filtering --min_qscore 10` Guppy option).

All basecalling operations were launched in GPU mode, using a Nvidia Tesla v100 GPU.

**Mapping reads to both strands of the reference.** After basecalling, reads were aligned
against their reference genome, using Minimap2 [16], with default parameters for ONT
sequencing, and options `--secondary = no --sam-hit-only` to discard unmapped
reads and secondary alignments (only the best alignment is kept in case of multiple alignment
positions in the reference). Moreover, reads ends are often "soft clipped" by the aligner, mean-
ing that these read portions do not map the reference. We discarded reads for which the total
length of soft clips represented more than half of the original read. For each bacterial dataset,
we found mostly less than ten occurrences of such highly soft clipped reads. It represents less
than 0.01% of reads (except for *S. marcescens* dataset for which it represented 5%; around 0.5%
for human datasets). These highly soft clipped regions did not show notably low quality scores,

but failed to be fully aligned. Moreover, we found that, although most of these large soft clipped regions that overlap do not align, some do. This suggests that these soft clipped regions might actually represent true variations from the reference genome.

The reference genome is represented by its forward strand sequence *G*. A given read *R* must be aligned either to *G*, or to its reverse-complement *rev*(*G*) if it is on the reverse strand. In the second case, most aligners use instead the reverse-complement *rev*(*R*) of the read in order to reduce the computational cost of indexing *rev*(*G*). However this raises two problems. First, given that the aligner uses heuristics, the alignments of *R* to *rev*(*G*) and *rev*(*R*) to *G* may differ. Second, it is important to keep the information of the sequenced read in order to correctly label the errors.

Considering a short example, with a read *R* = *GCCAAGACCT* aligned on the corresponding part of the genome *G* = *ACGTCATTGC*:

$$
\begin{array}{ll}
G \quad\; \texttt{ACGTCATTGC} & rev(G)\; \texttt{GCAATGACGT} \\
\quad\;\;\; \texttt{| ||| | ||} & \quad\quad\;\; \texttt{|| | ||| |} \\
rev(R)\; \texttt{AGGTCTTGGC} & R \quad\;\; \texttt{GCCAAGACCT}
\end{array}
$$

The left alignment represents what most alignment tools currently do: the read is reverse-complemented and aligned on the forward strand of the reference genome. On the right alignment is what we have chosen to do: the read is unchanged and the genome is reverse complemented. Here for simplicity we suppose the aligner performs an exact alignment and does not use heuristics. In both cases, the error rates are the same, *i.e.* 30% (they may actually differ due to aligner heuristics). However, the errors themselves are different: in the left case errors are *C/G*, *A/T* and *T/G*, while in the right case errors are *A/C*, *T/A* and *G/C* (*X/Y* denotes a substitution of a base *X* in the genome by a base *Y* in a read). Thus, the impact of the chosen alignment method for error profiling is crucial.

Therefore, for each set of reads, we run the aligner twice: against the reference genome, and against its reverse complement. In both cases, we forced the aligner to not reverse complement reads (`--for-only` option for minimap2). For each read, we kept the best alignment according to the forward or the reverse strand.

## Computation of quality score for sequenced reads

- **Phred quality score**. The standard quality score used by most sequencers is the Phred quality score *Q*, which is defined as $Q = -10 \times \log_{10}(P)$, where $P = 10^{\frac{-Q}{10}}$ is the sequencing error probability. For example, a Phred score of 10 is associated with a 10% error rate and a score of 20 with 1% error rate.

- **Nanopore quality score**. After Nanopore sequencing and basecalling, reads are gathered in fastq files that include a quality score for each base of each read. As for Illumina data, Nanopore quality scores are encoded with ASCII characters from 33 to 126 (the higher the value, the better the quality expected), representing Phred scores 0 to 93. In practice, the Nanopore score may differ from the Phred score (see Results), which to our knowledge is not reported anywhere.

## Computation of error rates

Error rates were computed as ratios of differences (insertions, deletions, mismatches) between each pair of aligned bases, over the total alignment length. The global error rate for a given

alignment was computed as the sum of all insertions, deletions and mismatches, over the total alignment length.

## Sequencing low complexity regions

We evaluated the sequencing accuracy of low complexity regions by measuring identity for different sequence lengths: this is the ratio of perfectly sequenced regions over the total number of sequenced regions.

- **Homopolymers**. A $k$-size homopolymer is a consecutive repetition of $k$ times the same base, with $k \geq 2$. We consider that a homopolymer is correctly sequenced in a given read if and only if it matches an identical homopolymer (same base, same length) in the reference genome.

- **Heteropolymers**. We call heteropolymer of length $2k$ a consecutive repetition of $k$ times the same couple of bases $XY$ with $X \neq Y$ and $k \geq 2$. We consider that a heteropolymer is correctly sequenced in a given read if and only if it matches an identical heteropolymer (same couple, same length) in the reference genome.

- **3-mers repeats**. 3-mers repeats, also called trinucleotide repeats, are made of triplets of three consecutive bases. 3-mers repeats of length $3k$ are consecutive repetitions of $k$ times the same base triplet ($XYZ$) such that $k \geq 2$ and $X = Y \Rightarrow Y \neq Z$ (homopolymers are excluded to remove their influence). We consider a 3-mer repeat to be correctly sequenced in a given read if and only if it matches an identical repeat (same triplet $XYZ$, same length) in the reference genome.

## Perfect k-mers

The NanoOK analysis tool [12] provides multiple statistics and figures to give insights into the sequencing quality of a given run. Among them, we retained the analysis of perfect k-mers, which are words of size $k$ that are sequenced in the run without any errors. This gives an interesting indication of the maximum resolution that can be expected from the technology without a read correction mechanism. In order to characterize the asymmetry of the longest perfect k-mers distribution, we used the skewness coefficients of Pearson (SCP) with respect to mode or median, which measure the difference in number of standard deviations between the mean and the mode or median. We use formulas $SCP_{mode} = (mean - mode)/stand.deviation$ and $SCP_{median} = 3(mean - median)/stand.deviation$.

## Sequence-specific errors

A certain type of sequencing errors, called sequence-specific errors (SSEs), are induced by particular sequences in the immediate environment of the error position. SSEs have been shown to exist in NGS data, with various patterns such as homopolymers [17] or inverted repeats [18].

We thus looked for *harmful k-mers*, words of size $k$ that are frequently associated with a given type of error. We fixed $k$ to 5, since for chemistry R9.4, this is the number of bases in the pore contributing to the signal. For each position of error and each possible word of size 5, we counted its number of occurrences in reads just before or after this position. We retain the 10 most frequent occurrences for each bacterial species (*i.e.* 120 words in total).

We used Weblogo [19] to represent the set of harmful k-mers. Logos show the letters that contribute most to the error at each position. Because they cannot account for global

properties such as base composition, as it has been observed in the case of GC rate (see Results), we completed this analysis by representing the set of harmful k-mers as a formal language. More specifically, we used finite automata, a graphical representation of regular expressions. A finite automaton is a graph where edges denote the reading of letters along the words, and the nodes (or states) correspond to the set of words read by following the paths from the initial node to this node. The words recognized by a finite automaton are those in distinguished nodes, called final states. We constructed automata for harmful k-mers right before and right after sequencing errors for indels and mismatches.

### Electrical signal from Fast5 files

We analysed the raw electrical signal from Fast5 files of our datasets, and sought for correlations between this signal and the observed errors, in particular exploiting the translocation speed.

We used the tool Tombo from ONT (https://github.com/nanoporetech/tombo), more precisely the "resquiggle" algorithm. In a nutshell, it uses the Fast5 files with added basecalled reads (from fastq files) as well as the associated reference genome, to map the signal to the reference genome.

Thus we get, for each reference base, an estimation of the signal, *i.e.* the number and the value (normalized by Tombo) of all sampled electrical current. Knowing the rate at which the electrical is sampled, 4.000 samples per second according to ONT, we computed the translocation rate, *i.e.* the speed at which bases go through the pore during the sequencing. Given a stretch of $N$ bases sequenced, a number of electrical signal $E$ sampled at rate $R$, the translocation speed of bases $S$ can be computed as: $S = \frac{N}{E} \times R$. For current chemistry, bases are supposed to translocate at an approximately constant rate of 450 bases per second.

## Results and discussion

### Guppy HAC basecalling mode reduces error rates by about 2% compared to FAST mode

We first evaluated the HAC and FAST Guppy basecalling modes, focusing on bacterial datasets. We provide results for both versions, 4.2.2 and 3.3.3, to show the software's evolution. FAST mode has not improved between the two versions and the difference in error rates between this mode and HAC mode has grown high enough for us to advise against continuing to use FAST mode, especially since the efficiency of HAC has improved and it is now only 2 (instead of 6) times slower than FAST in our measures.

For each studied species and both basecalling modes, we computed global error rates (sum of insertions, deletions and substitution errors) on reference-read alignments (Table 2). We also provide the median and mean error rate and runtime over all bacterial species. For HAC mode, the global error rate reaches about 5.7%, with some variations depending on the species (less than 2% magnitude). We got a similar profile for FAST mode, except that the error rate is about 2% higher. Error rates for human datasets are 6.6% with HAC and 8.5% with FAST.

In order to check for possible local effects, we also computed error rates (mismatches, insertions and deletions separately) on a sliding window along sequenced genomes (of size 1% the genome length). Fig 1 shows deletion errors, the most frequent ones, in the bacterial datasets using HAC. For each bacterial dataset, reads were aligned to the corresponding reference genome, then for each percent position in the reference genome we computed the mean error rate. The bacterial datasets are here represented as linearized genomes, with the start of the genome set to replication origin, located with Ori-Finder [20] from fasta files of reference

**Table 2. Global error rate and runtime for bacterial species, in FAST and HAC basecalling modes, comparing versions 3.3.3 and 4.2.2 of Guppy.**

| Species (strain) | Error rate (%) | | | | Runtime (min) | | | |
|---|---|---|---|---|---|---|---|---|
| | Guppy v4.2.2 | | Guppy v3.3.3 | | Guppy v4.2.2 | | Guppy v3.3.3 | |
| | FAST | HAC | FAST | HAC | FAST | HAC | FAST | HAC |
| *A. pittii* | 8.02 | 5.37 | 8.00 | 6.75 | **0.6** | **1.1** | **0.36** | **2.51** |
| *H. haemolyticus* | 7.24 | 5.72 | 6.97 | 6.00 | 0.73 | 1.36 | 0.48 | 2.53 |
| *K. pneumoniae* (INF032) | 8.21 | 6.23 | 8.63 | 7.41 | 2.61 | 5.65 | 1.54 | 10.0 |
| *K. pneumoniae* (INF042) | 8.10 | 6.11 | 8.32 | 7.14 | 2.89 | 6.61 | 1.65 | 11.92 |
| *K. pneumoniae* (KSB2 1B) | 8.17 | 6.24 | 8.46 | 7.29 | 2.39 | 5.09 | 1.39 | 8.94 |
| *K. pneumoniae* (NUH) | 8.85 | 6.63 | **9.14** | **7.95** | 1.73 | 3.69 | 1.08 | 6.44 |
| *S. marcescens* | 8.50 | 6.00 | 8.42 | 7.28 | 1.04 | 1.94 | 0.69 | 3.2 |
| *S. sonnei* | 7.98 | 6.15 | 8.13 | 7.24 | 2.8 | 5.49 | 1.68 | 9.19 |
| *S. aureus* | 6.78 | 4.27 | 6.71 | 5.53 | 1.26 | 2.71 | 0.78 | 4.78 |
| *S. maltophilia* | **8.87** | **6.83** | 8.69 | 7.63 | 2.99 | 6.68 | 1.76 | 11.97 |
| *S. thermophilus* (CNRZ1066) | **6.53** | **4.03** | **6.50** | **5.20** | **5.98** | **10.63** | **6.04** | **14.52** |
| *S. thermophilus* (LMD-9) | 7.06 | 4.65 | 6.95 | 5.68 | 4.41 | 7.91 | 4.43 | 12.09 |
| Mean | 7.86 | 5.69 | 7.91 | 6.76 | 2.45 | 4.91 | 1.82 | 8.17 |
| Median | 8.06 | 6.06 | 8.23 | 7.19 | 2.5 | 5.29 | 1.46 | 9.07 |

For each species and each basecalling mode, runtime is computed as the median time over five basecalling runs. Highest and lowest values of each column are in bold.

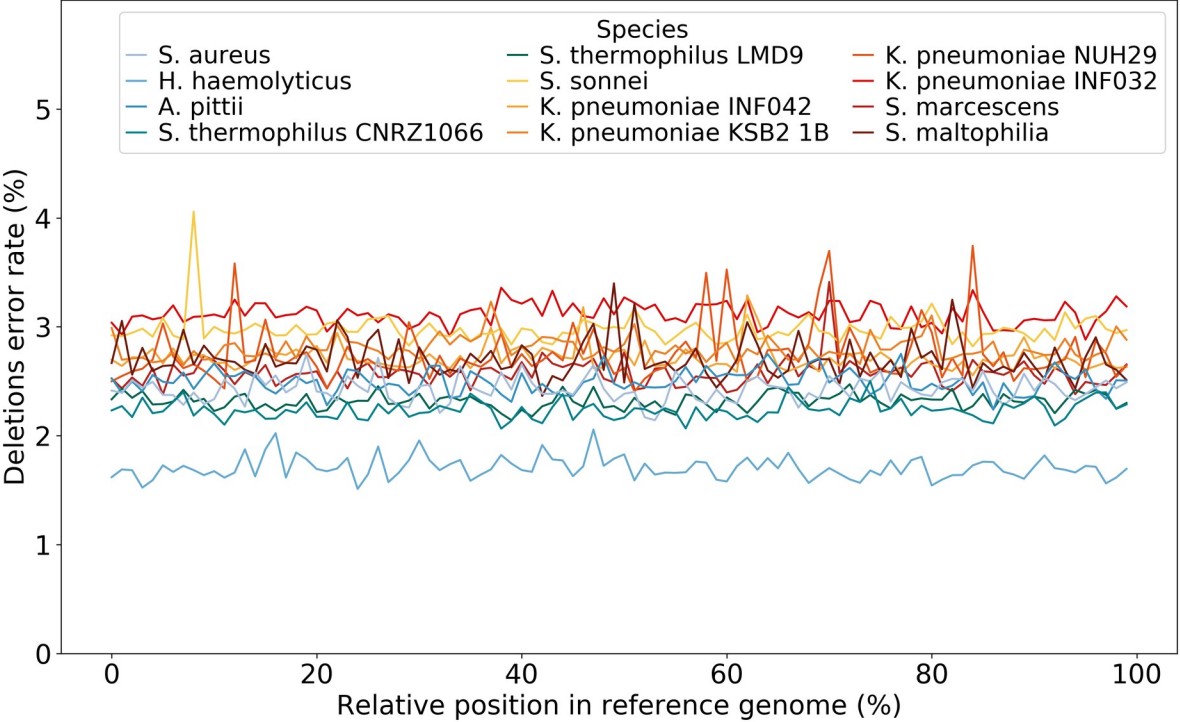

**Fig 1. Mean deletion error rates on bacterial genomes basecalled with HAC mode.** Species are sorted on their GC content, from blue (low) to red (high).

genomes. FAST basecalling mode showed similar results, but with slightly higher error rates (data not shown).

It appears that neither of the basecalling modes is biased according to genomic location, for each of the three types of error. Error profiles stay similar: most of the sequencing errors are deletions, followed by mismatches then insertions, which is consistent with a recent study [21]. For each type, the error rate is quite constant along genomes (with some possible local peaks): within 1.6–2.7% for deletions, 1.2–2.2% for mismatches and 1.1–2.4% for insertions. The error variation is not high (standard deviation around 0.1% for deletions). Even for peaks, the difference hardly exceeds 1%. Nevertheless, we have studied the sequences for peaks above 3 standard deviation around the median. These sequences present no particular pattern, neither from the point of view of complexity, GC content, or type of annotation (coding mostly for proteins).

An analysis of raw electrical signal associated to deletion of at least 5 bases showed that the translocation rate was at least twice as fast on deleted parts (median $\simeq$ 1,220 bases per second) than right before (median $\simeq$ 460 bases per second) or right after (median $\simeq$ 520 bases per second) these errors. We thus computed the error rate of reads depending on translocation speed (section "Raw signal analysis" in supplementary data, S10 Fig). While the current expected speed (around 450 bases per second) is associated to low error rates (median below 5%), a higher translocation speed lead to a dramatic increase of error rate (above 20% for speeds over 650 bases per second). As mentioned above, it is to be linked with deletion errors. More surprisingly, a slow base translocation is also associated to higher error rate (5 to 10% for speeds below 150 bases per second), this time being mainly associated to insertion errors.

For RNA datasets, the global error rate is higher and reaches around 9.65%. The distribution of error types also differs: there are more deletion errors (51%), less mismatches (27%), and a similar insertion rate (22%) for RNA sequencing data. The mean error rates for FAST are around 0.7% higher than the mean error rates for HAC. Thus HAC mode is particularly useful. In the previous version of Guppy v3.3.3, these differences were not as strong: FAST error rates were about 0.4% higher.

It is worth noting that quality of sequenced raw reads drops at both ends (S1 Fig), around 100 bases, including adapters and part of the DNA/RNA sequence. These lower quality scores are related to the initiation and termination fragments of the nanopore measurements, for which the signal is probably less stable. These error-prone read sections are generally clipped by the aligner (on average, the first and last 130 bases). For RNA datasets, the quality drop at read ends concerns globally the first 10 bases and last 20 bases, which are also clipped by the aligner.

Overall, the global error rate difference is not the sole argument for the genuine improvement represented by HAC mode. For more specific types of errors the improvement may be huge, exceeding 80% for instance for correctly sequenced homopolymers of size 6, which represent more than half of the homopolymers of this size for HAC mode (S2 Fig). All results in the following were obtained using the HAC basecalling mode.

## Bias in substitution errors

For genotyping of individuals, the study of Single Nucleotide Polymorphism is of great importance and Nanopore sequencing is a promising technology for applications such as forensic profiling [22], although it is still imperfect, presenting a higher degree of mismatches than NGS. More detailed knowledge of this type of sequencing errors is essential to help reduce the signal-to-noise ratio between true variations and Nanopore sequencing errors.

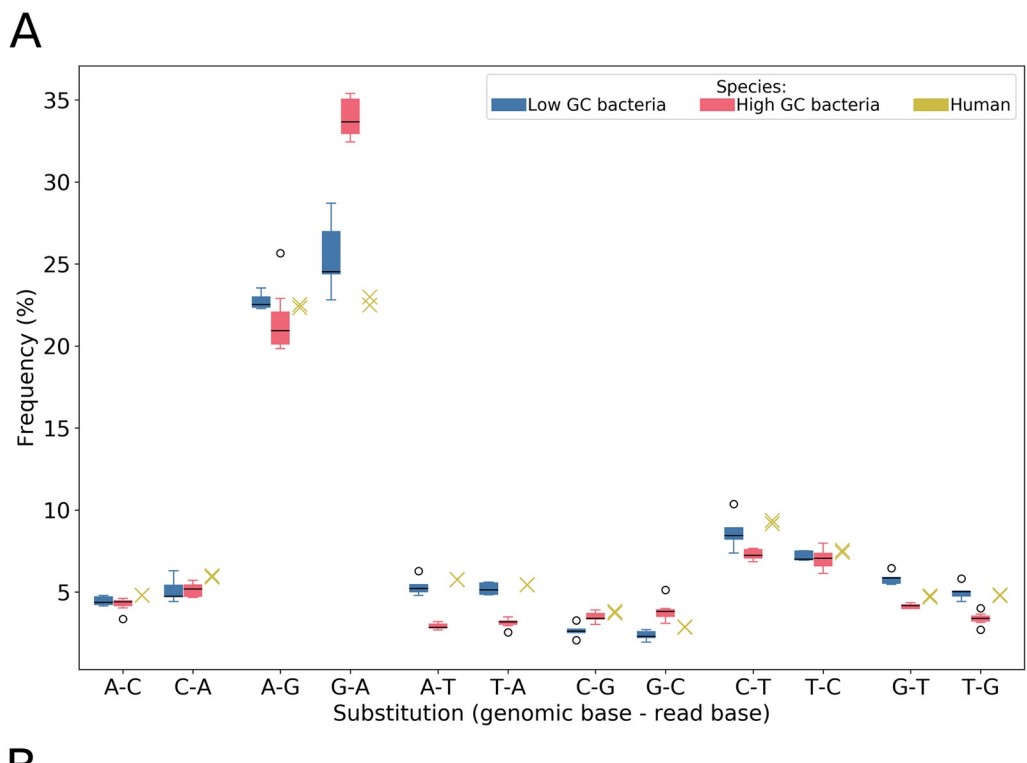

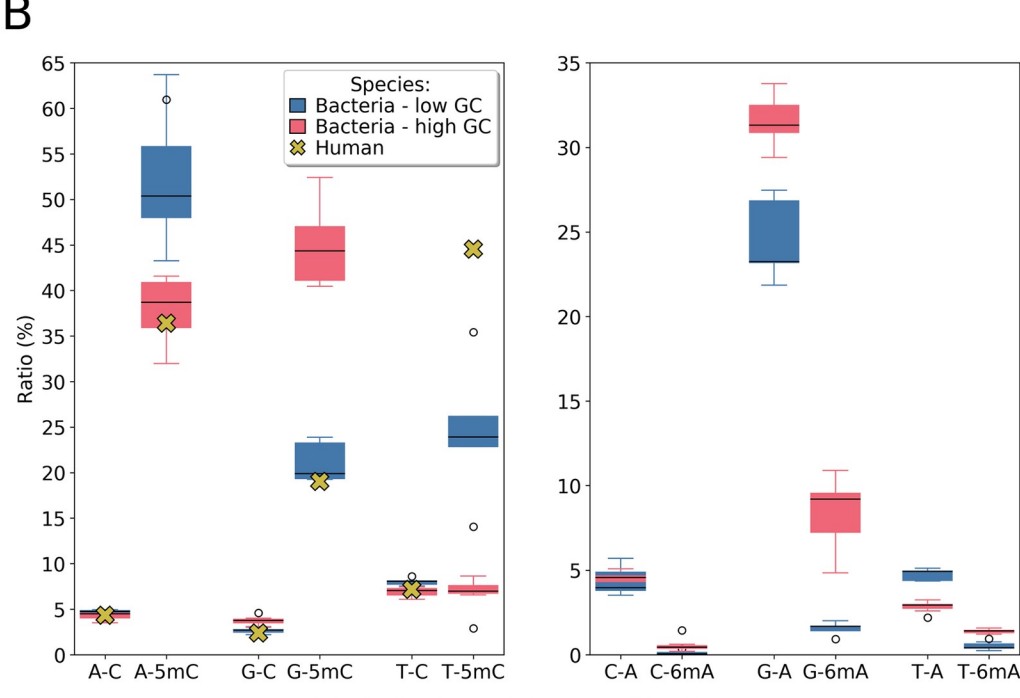

**Fig 2. Substitution error profile for bacterial and human datasets.** Results for bacterial datasets are presented with boxplots, those for human are displayed with crosses. **A**: Results obtained in HAC basecaller mode (unaware of methylation). **B**: Results obtained in methylation basecaller mode. We focused here on substitution to A and C in reads, the only methylations supported by Guppy (only the C-methylation for human data). For each substitution type, its abundance is given relatively to all substitutions, separating methylated and non-methylated positions.

**Transitions are more frequent than transversions.** For each dataset studied, we enumerated the twelve possible pairs of substitutions $X - Y, X \neq Y$, where $X$ is the nucleotide in the reference genome and $Y$ the corresponding nucleotide in the read (Fig 2A). We found that, for bacteria as well as for the human dataset, substitutions between A and G occurred about three to five times more often than the transversion error. We also noted that substitution between C and T bases were also a bit more frequent than transversions. We performed two t-tests (one comparing A-G/G-A substitutions, the other comparing C-T/T-C substitutions against transversions) and found that transitions were significantly more frequent than transversions (p-value = 1.08e-78 and 1.48e-31 respectively).

This bias is likely due to the internal chemical similarities within the groups of purines (A and G), and pyrimidines (C and T), which are much higher than inter-group similarities. As a consequence, the electrical signals are harder to discriminate between two purines (two-ringed structures) or two pyrimidines (one ring). A similar phenomenon was described for human data [23] and for *E. coli* data [21], where authors found that transitions (and not only substitutions between A and G) were about three times more frequent than transversions. We found very close results if we did not apply the specific handling of reverse complement mapping reads (see "Mapping reads to both strands of the reference" subsection of Materials and Methods). Thus we suspect the discrepancy compared to our findings might be due to some incorrect handling of reverse-mapping reads.

**The substitution error profile changes for methylated bases, depending on the GC rate.** We repeated a similar analysis, this time taking methylation prediction information into account. Our aim was to determine whether the methylation of certain bases has an impact on the substitution error bias. We separated the bacterial results from the human ones, as different methylation types are involved (6mA dam and 5mC dcm methylation for bacteria data, 5mC CpG methylation for human). After basecalling each dataset in a dedicated methylation mode, reads were aligned against the associated reference genome, and substitution counts were finally established with respect to the predicted methylation status of the base in the read.

We compared the substitution error distribution in two cases (Fig 2B):

(1) considering that a cytosine or adenine appears in a normal state in the read, and (2) considering that the substituted base appears methylated in the read.

The substitution error profile is quite similar between bacteria and humans for non-methylated bases. The global trend of over-represented transition substitutions is found again for non-methylated bases, but not for methylated ones. Moreover, for methylated bases, the error profile is quite different depending on whether one considers low- or high-GC bacteria. Low-GC bacteria have barely any substitution to methylated adenine, whereas high-GC bacteria display a profile that is closer to that for non-methylated bases.

For methylated cytosine, the difference is more pronounced with respect to unmodified ones. It also appears that despite quite even number of predicted methylated bases between 6mA and 5mC (same order of magnitude, around $10^6$ for bacteria), the ratio of predicted 6mA involved in a substitution is about tenfold lower than the ratio for 5mC.

The treatment of methylations seems to have been one area where recent basecallers have improved [4]. These results seem to show that there is still room for improvement with respect to the prediction of methylated C.

## Low-GC species and reads are globally better sequenced

NGS-sequenced data are known to exhibit GC-bias, mostly due to PCR amplification [24, 25], causing either GC-rich or AT-rich DNA sequences to have a lower depth of sequencing.

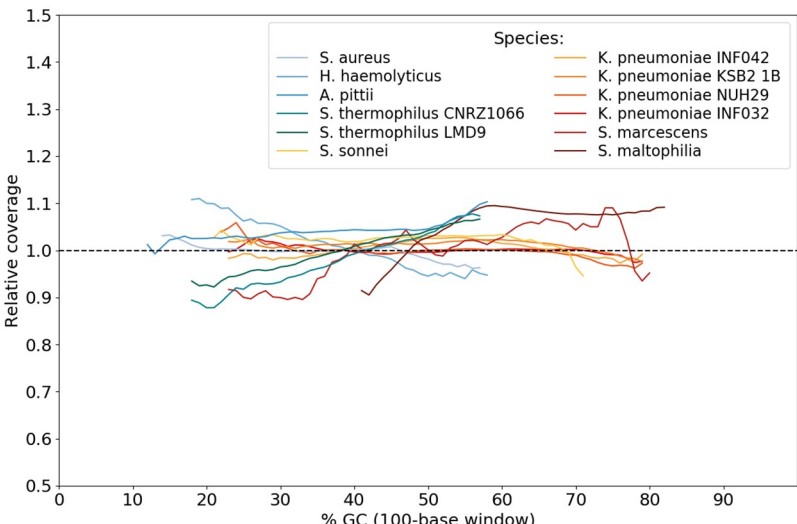

**Fig 3. Relative coverage for each bacterial species, according to local GC content.** The black dotted line represents a GC-unbiased depth of coverage. Only points associated with at least 1000 windows in the genome were plotted.

Several articles [26–28] have concluded that Nanopore sequencers do not suffer from such bias, as its library preparation does not require a PCR amplification step.

In fact, GC content bias is almost always considered in terms of *read coverage*, but there are other features worth considering. Here we analyze how the GC rate affects sequencing from several perspectives, be it in terms of depth of coverage, quality and/or error rate.

We first checked that Nanopore sequencing was indeed free from a bias in depth of coverage. For each bacterial dataset separately, we computed the relative coverage for various %GC values of the genome. The relative coverage [29] is normally defined as the coverage of a given reference base in a genome divided by the mean coverage of all reference bases. Here, we consider instead 100-bases windows. Results are summarized in Fig 3.

Globally, depth of coverage is not highly impacted by the GC content of the sequence. This general result must, however, be considered with some caution because we observed for two species a quasi-linear sensitivity of the sequencing depth to the GC rate. This occurs for species *S. thermophilus* and *H. haemolyticus*, in opposite directions.

We then computed error rates (mismatches, insertions, deletions and total) as well as GC content for each read, regardless of the species (Fig 4). For bacteria (Fig 4A), the curves are almost flat apart from a slight increase in the mismatch rate with respect to the GC content. However, if one distinguishes the species with a low GC content (below 50%) from those with high GC content (above 50%), then a more pronounced discrepancy appears (dotted lines, about 1.5% error shift), suggesting a more global effect of the GC rate on the error level. For human reads (Fig 4B), the error increase relative to the GC rate increase is much more pronounced. Results for Guppy v3.3.3 were similar, although the global error rate was about 1% higher.

Interestingly, the same trend can be observed through quality (a measure that we will consider in depth in the next section). For this purpose, we computed the quality of sliding windows of size 100 nucleotides, as a function of their GC content on bacterial reads (S3 Fig). A curious drop in quality appears around the central 50% GC value, which may be the trace of a hidden adjustment threshold for quality calculation since it was absent from version 3.3.3. On either side of this threshold, quality and GC rates are inversely correlated.

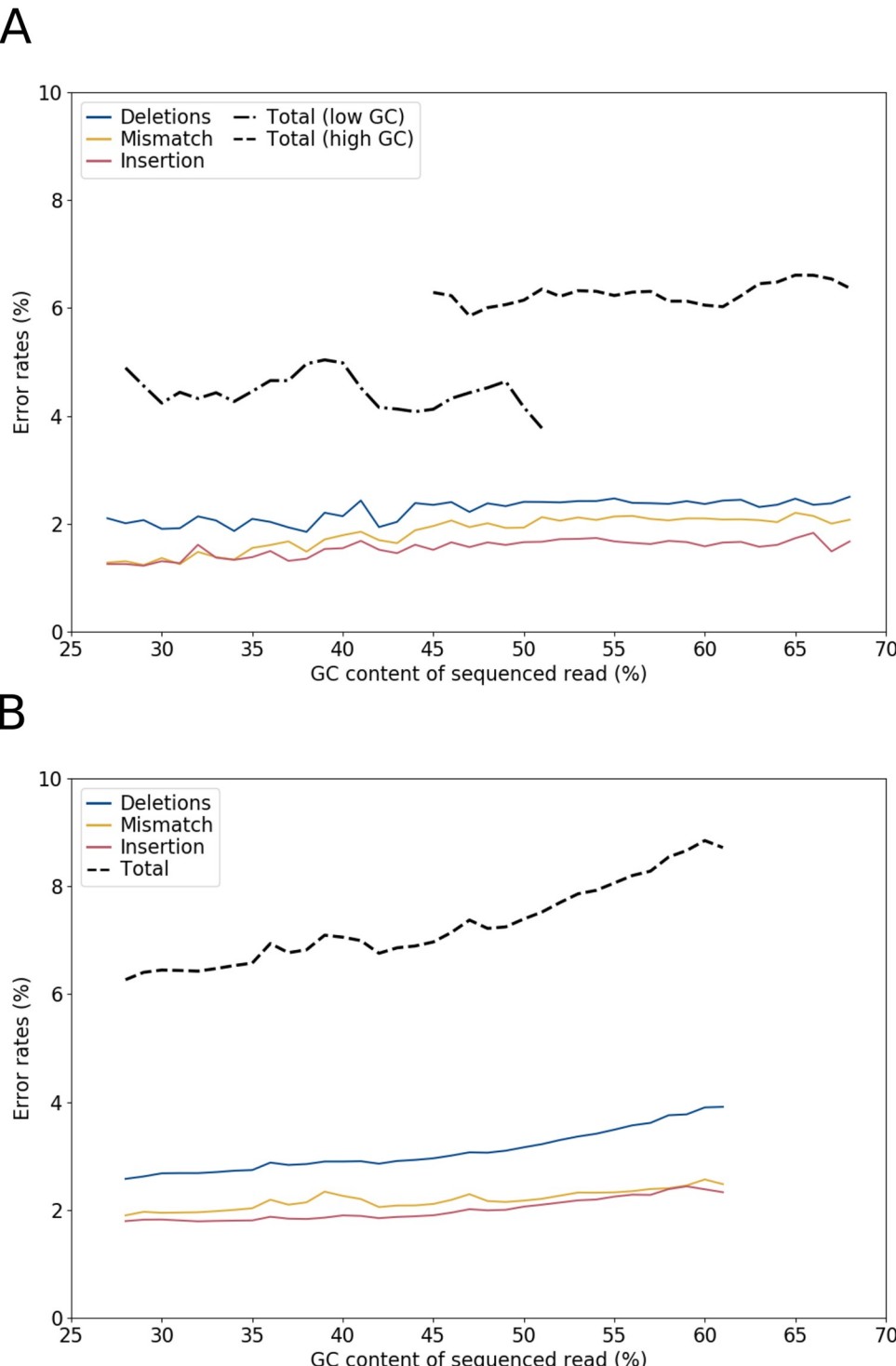

**Fig 4. Global error rates of reads according to their GC content.** GC contents were grouped by integer value. Only values supported by at least *n* reads were plotted. For each species the mean error rate is computed, and the median on all this values is displayed. **A**: Results for bacterial data, *n* = 100. Low-GC species have lower global error rates than high-GC ones. **B**: Results for human data, *n* = 1,000. Error rates show a correlation with the GC content of reads.

This indicates that GC content directly affects how well the read is sequenced. Note that the sensitivity of sequencers to GC content has also been observed for NGS devices [30]. For instance the Miseq system has a substitution error rate that increases by as much as 2% for sequences with high GC content ($> 60\%$). For the MinION device, we observed a relationship for all types of errors.

## Nanopore quality scores do not follow Phred scores, yet can be used to estimate error rates

The knowledge of the error rate in the reads is fundamental for all the processing and analysis that will be performed later on these data. It is easy to estimate by mapping them on a reference genome. However, the reference genome can be of poor quality, represent a different strain, or even be absent. Thus, being able to evaluate the error rate of reads without reference is an objective of primary importance. The quality index associated with sequences is generally conceived as a simple criterion for filtering the reads, but one can also question in a much finer way whether it is adapted to the estimation of the error rate.

We exploited the quality scores provided in fastq files. For each sequenced read of studied species, we computed its error rate on 100-bases sliding windows, depending on its associated mean quality score (see Fig 5). First, we observed that for our data, for the most frequent quality score values (*i.e.* between 7 and 30), quality scores do not match expected Phred scores:

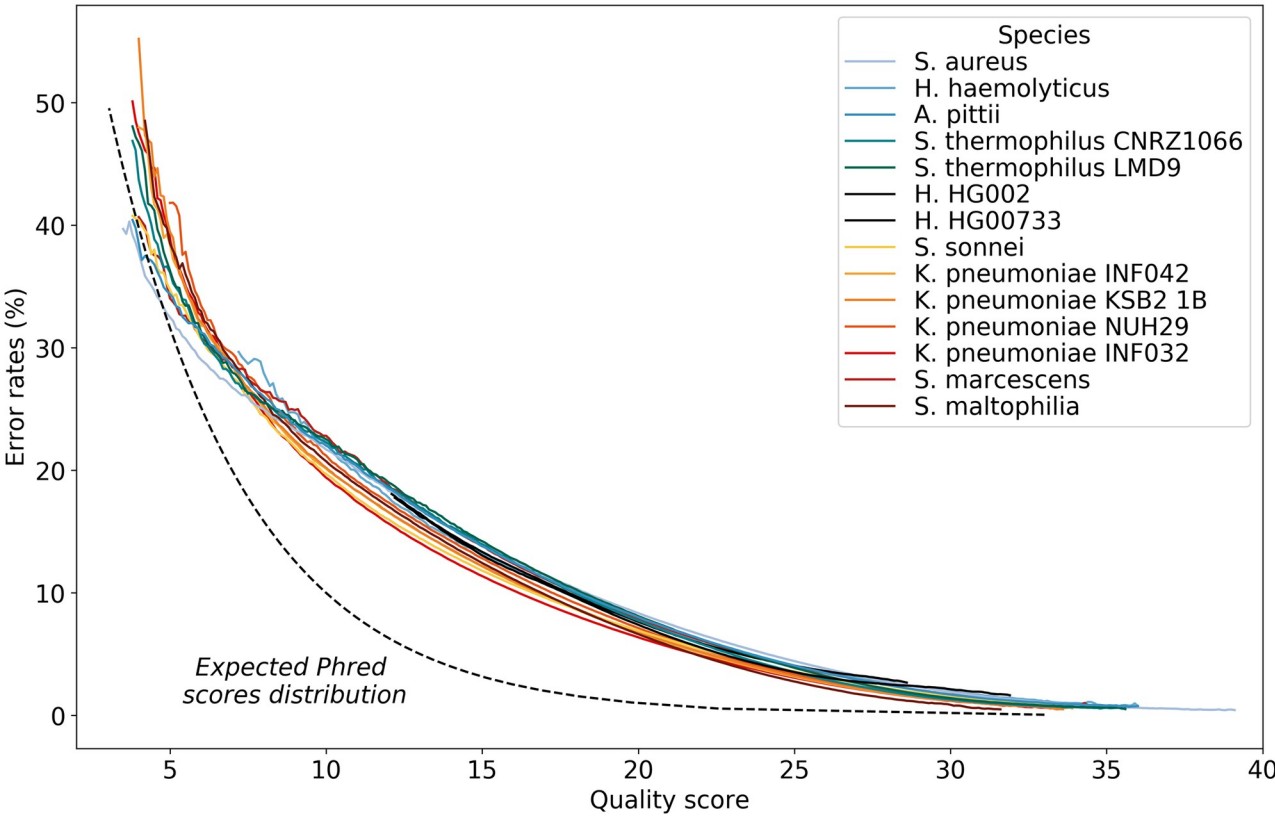

**Fig 5. Mean error rate depending on quality score for sliding window along reads.** Sliding windows are of size 100. Quality scores are rounded to the first decimal value. The dotted black line represents the expected Phred score relationship between quality score and error rate, other lines represent results obtained for our studied species. Results were computed on all bacterial aligned reads, and on 100,000 aligned reads for each human dataset. Only values supported by at last *n* reads are shown (*n* = 10 for bacterial data, *n* = 10, 000 for human data).

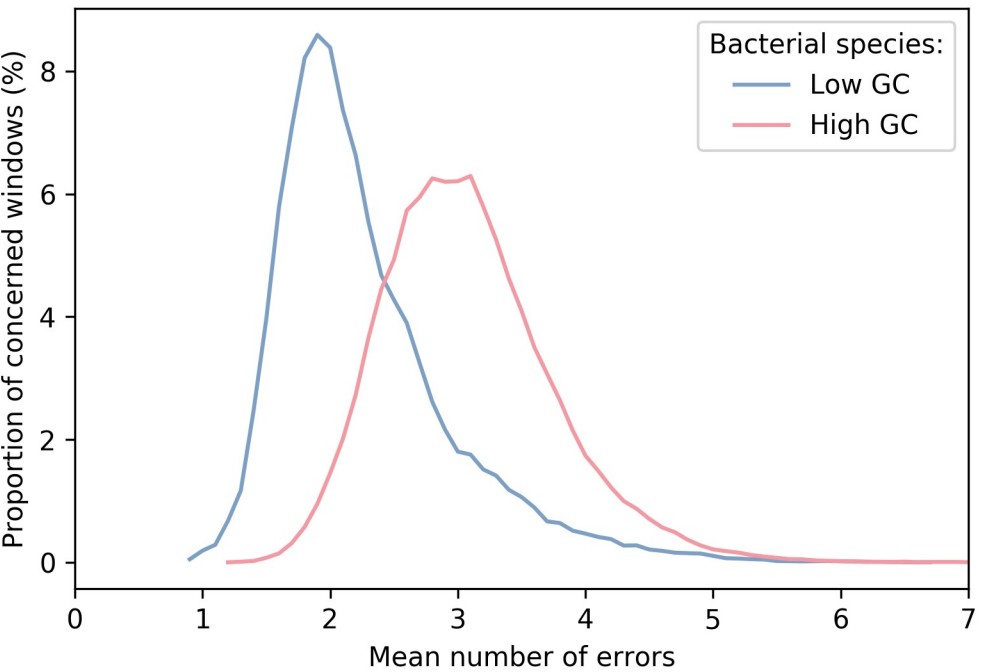

**Fig 6. Distribution of read windows according to their error rate.** For each bacterial species, we computed the mean number of errors (rounded to the decimal) of read parts that align to windows of size 50 in the reference genome. We expressed the number of errors as the median number of errors for each window and then plotted the global window distribution for both low- and high-GC content species.

they are clearly overevaluated. Moreover, the curve is more linear than what is expected with Phred score distribution.

We computed a similar graph, at read scale (S4 Fig) and found same results, with the difference that the range of quality score values is smaller. This was already noted in [31], although quality scores are now much closer to Phred scores than they were then.

The good news is that Nanopore quality scores are correlated with error rates, and this consistently for all studied species. This dependency is quasi-linear with a slight quadratic trend. Overall, the error rate $E$ depends on the quality score $Q$ following approximately equation $E = 0.042Q^2 - 2.68Q + 43.92$ ($R^2 = 0.99$) in the acceptable range $Q \in [7, 30]$.

## Correlation between error rate and amount of reads

Another useful prediction is the proportion of reads that can be expected at a given level of error or quality. Nanopore recommends a minimum threshold of 7 for data quality. We computed the distribution of reads according to their error rate, for thresholds ranging from 7 to 12, for bacterial datasets (S5 Fig). We decided to set the threshold for data quality to 10, as this yields more than 98% of reads below the 10% error rate (in contrast, the minimum threshold retains nearly a quarter of reads with an error rate greater than 10%). We estimated on bacterial datasets that this more realistic threshold corresponds to a more predictable error rate and reduces it globally by about 0.5 to 2.5% compared to the minimum threshold (see Fig 5).

The counterpart is that this leads to a reduction in the number of reads (ranging from 5% to 33% for the most extreme dataset *K. pneumoniae* KSB2 1B). In fact, between values 7 and 10, the error rate and the amount of reads at this rate are clearly correlated (S6 Fig). It follows a

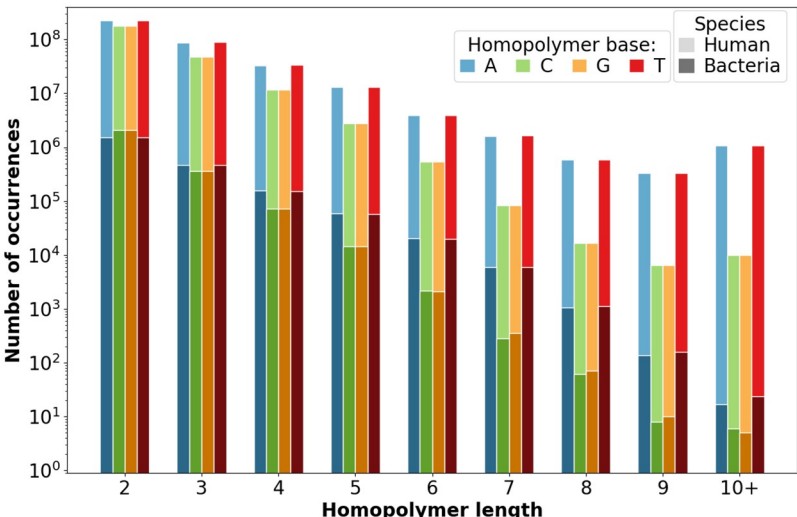

**Fig 7. Homopolymer length distribution in bacterial (dark, bottom) and human reference (light, above) genomes.**
The scale is semi-logarithmic. Occurrences are summed within each category.

regression line of approximate equation $y = 13x - 0.4$ ($R^2 = 0.91$), $y$ being the ratio of reads and $x$ the ratio of error rate between thresholds 7 and 10.

More generally, the distribution of reads according to the error rate (or, equivalently, the quality) is the purpose of Fig 6. Again, we can notice around 1.5–2% error difference between modes for high- and low-GC bacterial species. Moreover, the results for high-GC species show a more variable but more symmetrical distribution than for low-GC species. The results for Guppy 3.3.3 were similar, with a slight error increase.

### Sequencing errors in homopolymers

Nanopore sequencers are known to struggle to accurately sequence low-complexity regions, such as homopolymers. We studied the sequencing errors associated with the homopolymeric regions of the reference genomes, and tried to delineate the contexts in which they are most frequent.

**Homopolymer distribution in genomes.** The distribution of homopolymer lengths for bacterial and human reference genomes (according to fasta file, representing the forward strand only) is shown in Fig 7 for the four nucleic acids. This gives insights on abundance of each homopolymer categories, before we investigate on their error-prone behavior in the following subsections.

In all cases, the global homopolymer distribution tends to decrease exponentially. For very short homopolymers there are no strong differences between nucleic acids, whereas for homopolymers exceeding three bases, G- or C-based homopolymers rapidly become scarce. This difference is about a ten-fold decrease, and can even reach two orders of magnitude for very long human homopolymers (ten bases and more).

**Nearly half of sequencing errors are due to homopolymers.** We evaluated the amount of sequencing errors due to homopolymers. To this end we browsed all alignments of reads on bacterial and human reference genomes and marked those in homopolymeric genomic regions (Table 3 for bacteria and Table 4 for human).

The results are similar for bacterial and human species: about 25–30% of insertion errors and more than half of mismatches and deletion errors are linked to homopolymers.

**Table 3. Bacteria: Total and homopolymer-induced sequencing errors.**

|  | Mismatches | Insertions | Deletions | Global |
|---|---|---|---|---|
| H: Homopol. ($\times 10^6$) | 31 | 13 | 41 | 84 |
| A: All errors ($\times 10^6$) | 59 | 50 | 74 | 184 |
| Ratio H/A (%) | 51.85 | 25.22 | 54.95 | 45.78 |

**Table 4. Human: Total and homopolymer-induced sequencing errors.**

|  | Mismatches | Insertions | Deletions | Global |
|---|---|---|---|---|
| H: Homopol. ($\times 10^6$) | 1, 027 | 533 | 1, 428 | 2, 988 |
| A: All errors ($\times 10^6$) | 1, 883 | 1, 709 | 2, 536 | 6, 128 |
| Ratio H/A (%) | 54.54 | 31.19 | 56.31 | 48.76 |

Altogether, about 47% of errors are due to homopolymers. Version 4.2.2 of Guppy lowered the number of insertion errors in homopolymers, at the cost of more deletions.

**Homopolymer sequencing accuracy depends on length and GC content.** The Fig 8 shows the rate of correctly sequenced homopolymers as a function of their expected length in the genomes. Homopolymer sequencing remains rather good ($>$ 70% errorless) until a length of 5 (4 for human and high GC bacteria). Above this length, the rate of well-sequenced homopolymers drops drastically (*e.g.*, only one fourth of homopolymers of length 8 is correctly sequenced). As expected, results are better for low-GC species. In particular, for lengths over 4, the worst results are obtained for the highest GC content. Guppy has evolved well in this respect since version 3.3.3, which had more problems finding homopolymers longer than 4 (for example, only about 30% of homopolymers of length 7 were correctly sequenced for low-GC bacteria, against more than half for the current version 4.2.2).

We also computed a detailed version of this figure, for each base (S7 Fig). For short homopolymers of length 2, the base of the homopolymer does not strongly influence the ratio of correctly sequenced homopolymers. However, for longer homopolymers, C and G

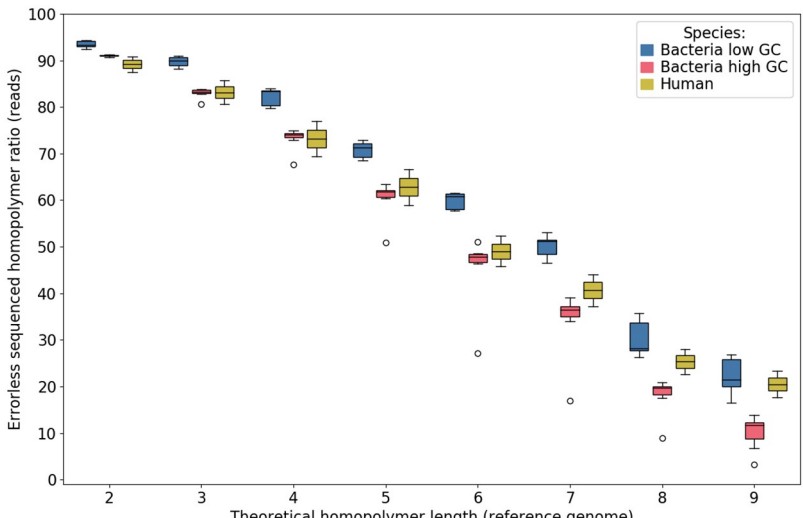

**Fig 8. Ratio of well-sequenced homopolymers as a function of their reference length.** Results are pooled into three categories: low GC (blue) and high GC (red) bacterial species, and human data (yellow).

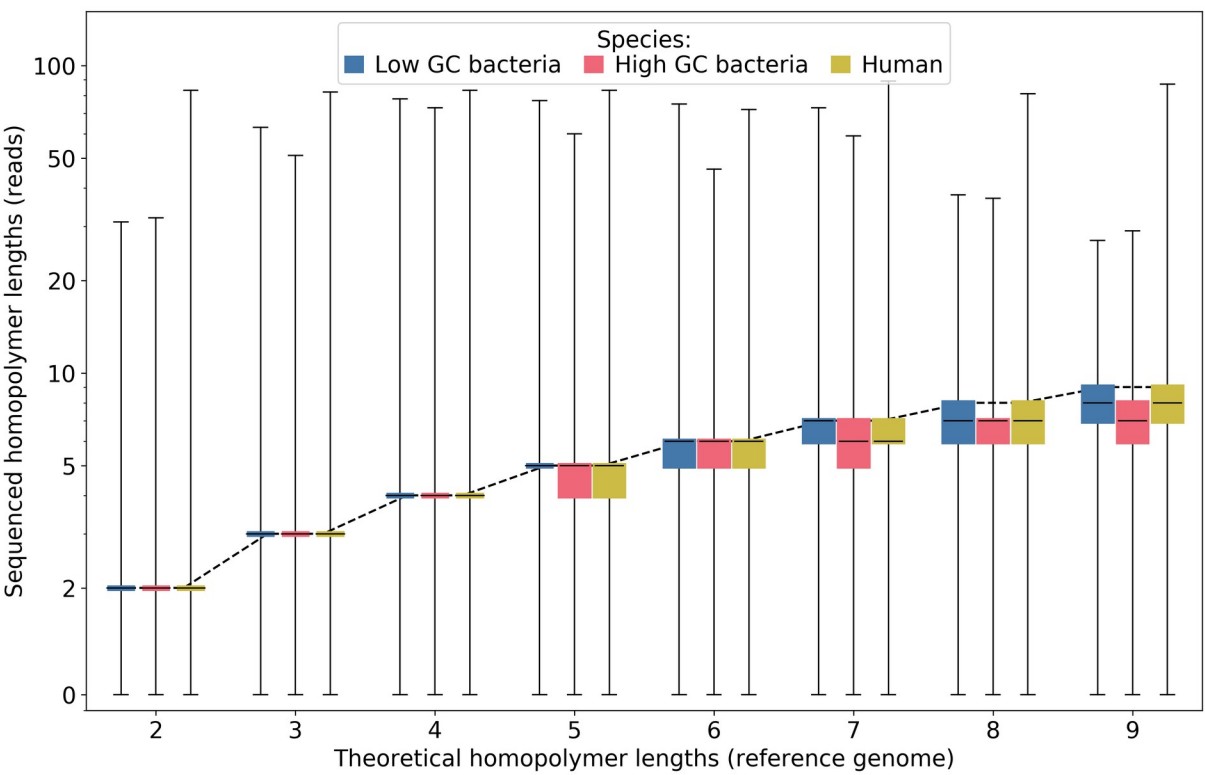

**Fig 9. Errors in sequenced homopolymer lengths.** The dotted line represents the expected length. The scale is semi-logarithmic.

homopolymers are significantly worse sequenced than A and T ones. This trend intensifies with the increase of homopolymer length. Results are consistent whether one considers low-, high-GC species and human.

It is known that the main source of errors on homopolymers with Nanopore technology is the estimation of their length. We therefore wondered if we could more accurately characterize this deviation from reality. Differences between sequenced and expected homopolymer lengths are shown in Fig 9. As previously seen, homopolymer lengths below 5 (or equal to 5 for low-GC species) are quite well estimated most of the time (narrow box). For longer homopolymers, length tends to be slightly underestimated. Another lesson is provided by the great variability observed for some sequences: in particular, there can be largely overestimated lengths, even for small reference sizes (one or two order of magnitude for homopolymers of size 2–5).

Compared to Guppy version 3.3.3, recognition of long homopolymers has improved and the overestimation of some homopolymers of length 2 slightly increased.

When considering the electrical current associated to homopolymer sequencing, one can observe two phenomena (section "Raw signal analysis" in supplementary data, S11 Fig). First (A and B figures), our results show that the electrical signal is highly impacted by the surrounding bases, as stated by Nanopore. In the context of homopolymers, this results into (1) an usually well segmented signal, both in terms of duration and value, for homopolymers of length ≤ 5 bases long, thus being mostly well sequenced, unlike (2) longer homopolymers for which the current is far less influenced by surrounding bases, which results in a harder to segment signal, making it harder to correctly assess the homopolymer length. Moreover (A and C

figures), it seems that the level of the electrical current is a quite good indicator of the sequenced base. It appears that signal values can be roughly ordered along the sequence in the following way: $signal(T) \geq signal(C) \geq signal(A) \geq signal(G)$. Clearly, it will be harder to distinguish for instance a transition A-G than a transition A-T. In practice, this also also impacts the way homopolymers are sequenced: an homopolymer of A, surrounded by T bases, will be better sequenced than an homopolymer of A surrounded by G bases, as the difference in signal level is more pronounced in the first case.

We end this section with details about error ratios in homopolymers for all types of errors (mismatches, insertions and deletions), shown in Fig 10.

As expected, most mismatches are found in short homopolymers. The deletion error rate rises and becomes predominant with the length of the homopolymers, a finding in perfect agreement with the sequencer's tendency to underestimate their size. Overall, profiles for human and bacterial data are similar. A notable exception is the rate of insertions in long homopolymers, which is significantly higher for human data. It reflects the existence of sequences whose size is largely overestimated. Compared to Guppy version 3.3.3, we noticed fewer insertions for small homopolymers (length $\leq 4$), and more deletions for longer ones.

## STRs do not escape the increased sequencing error rate

We also investigated a more general pattern of low complexity regions, Short Tandem Repeats (STR or microsatellites), which are stretches of short motifs (a few base pairs) that are relatively frequent in eukaryotes (*e.g.* about 3% for humans) but also present in prokaryotes. They are known to be challenging for aligners that have to choose between multiple very close solutions to match them and can produce many artefactual hits. This has an impact in particular on correction algorithms, which are generally hampered by the presence of these regions [32]. STR are generally polymorphic and cannot be simply filtered from sequences because of their

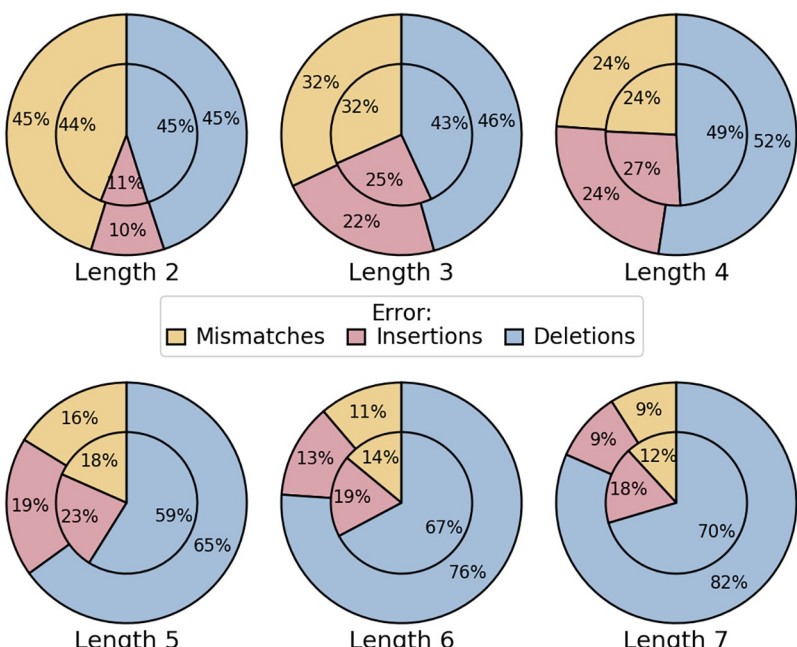

**Fig 10. Error rate ratios by type in homopolymers of various lengths.** For each length, the error ratios for each type (mismatches, insertions and deletions) are shown, for both bacterial (outside pie) and human (inside pie) datasets.

importance in terms of genetic markers (forensic applications) and perhaps even their functional role in regulation [33]. Despite this interest, little is known about the correction of their sequencing.

We started by a study of STR we called *heteropolymers*, made of dinucleotide motifs (see the distribution of their lengths in reference genomes in S8 Fig). Most heteropolymers are of length 4–6 (2 or 3 copies), and very few of them exceed 10 bases long. CG dinucleotides are an order of magnitude less frequent than the others.

As for homopolymers, differences between sequenced and expected heteropolymer lengths were computed (S9 Fig). Interestingly, we observed the same trend: a correct estimation for short heteropolymers (length 4–6), the size of longer ones being underestimated, and largely overestimated lengths for some short heteropolymer sequences.

Finally, we assessed sequencing error rates and the abundance of each type of heteropolymer, grouping species by their GC rate (see Fig 11). Error rates were similar for Guppy 3.3.3, although about 0.5% higher. Dinucleotide errors are uniformly distributed but can vary depending on the species considered and its GC content. Overall, the error rate is the lowest for low-GC species, and human data has similar profile to high-GC bacterial species, except for CG heteropolymers for which the error rate is much higher. It seemed that this higher error rate could not be related to CpG regions. Indeed, when we looked at predicted CpG regions (based on UCSC database, see Data Availability section), we found that the error rates of reads parts that aligned to these regions were very close to the global error rates of all human reads (*i.e.* around 6–8%), and not as high as the 11% observed for CG heteropolymers.

We also studied sequencing accuracy for another type of STR, trinucleotide repetitions (2, 3, and 4+ occurrences). Trinucleotide repeat expansion is a kind of mutation known to be involved in several more or less severe disorders (e.g. myotonic dystrophy or Huntington's disease [34]). For the analysis, nucleotides were merged in two groups of complementary base pairs: *S* (Strong) and *W* (Weak). We then defined four 3-mers groups, excluding already studied homopolymeric patterns: (1) *S*-only trinucleotides (*i.e.SSS* patterns), (2) mostly-*S* trinucleotides (*i.e.SSW*, *SWS* and *WSS* patterns), (3) mostly-*W* trinucleotides (*i.e.WWS*, *WSW* and

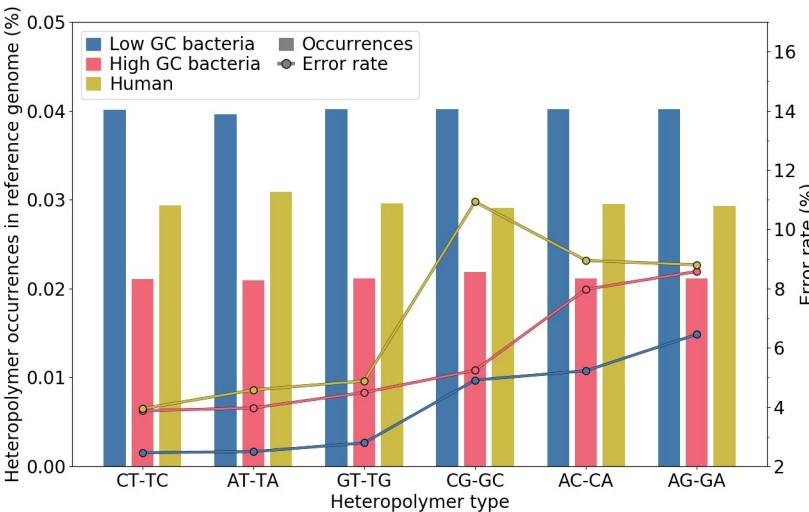

**Fig 11. Heteropolymer genomic mean abundance and sequencing error rates.** Species were grouped into three categories: low-, high-GC content, and human. Abundance (left axis) is represented with barplots, and error rate (right axis) with lines.

*SWW* patterns), and (4) *W*-only trinucleotides (*i.e.WWW* patterns). For instance, the fourth group gathers the following set of trinucleotides: {AAT, ATA, TAA, ATT, TAT, TTA}. The results are presented in Fig 12.

The distribution of occurrences (Fig 12A) is quite similar between the different repetition lengths, and trinucleotide repeats are rapidly becoming scarce as their length increases. For each category (low-GC bacteria, high-GC bacteria and human), frequencies are quite even, except for S-only trinucleotides, less abundant in low-GC bacteria and human datasets, and W-only, less abundant in high-GC bacteria. Low- (resp. high-) GC species have less occurrences of S-only and mainly-S (resp. W-only and mainly-W) trinucleotides, a trend that intensifies with repeat length.

Sequencing accuracy results are shown in Fig 12B. Unexpectedly, accuracy for bacterial datasets seems normal, quite constant for the different repeat lengths and close to the global error rates (error rate around 4–6%). For human datasets, trinucleotide repeats are worse sequenced than for bacteria, especially "S-only" and "mainly-S" ones.

Overall, for all species, trinucleotides containing more W bases are better sequenced, denoting again a GC bias in sequencing errors. However, one should keep in mind that larger datasets with high proportion of trinucleotides would be necessary to build a more precise profile: the number of occurrences of trinucleotides of size 4 and more is quite low here.

## Perfect k-mers

Perfectly sequenced k-mers are an interesting concept introduced in NanoOK [12]. It gives an insight into the actual size of the reads that can be trusted to resolve ambiguities during assembly. For each bacterial dataset, we looked for the longest perfect k-mer in each read, and computed two figures: the ratio of reads in which a perfect k-mer of size at least $k$ is present (Fig 13A), and the size distribution of the largest window without sequencing errors in a read (Fig 13B).

Both curves show a clear shape. Fig 13A is a sigmoid that is similar for low and high-GC bacteria, with a clear drop between values 150 and 300 for $k$. Fig 13B corresponds globally to a lognormal distribution of mode 200 with a slightly positive skew. This is compatible with a strong random component in the distribution of errors. Note that the expected distribution of errors under the assumption of randomness is more similar to a step function than to a sigmoid one, highlighting a residual probability of long perfectly sequenced k-mers that is greater than expected. The asymmetry of the distribution depends on the GC content. For low-GC bacteria, which are the best sequenced, the mode is 212, the mean 242, the median 487 and the standard deviation 242, corresponding to Pearson skewness $SCP_{mode} = 0.12$ and $SCP_{median} = -3$. For high-GC bacteria, the mode is 198, the mean 224, the median 353 and the standard deviation 224, corresponding to Pearson skewness $SCP_{mode} = 0.12$ and $SCP_{median} = -1.7$. In practice, the longest perfect k-mer for low-GC bacteria reads may be very long: for about one fifth of the reads it reaches 350 bases. Most mapping algorithms are based on the search of seeds in sequences, that is, substrings that mapped a reference without any error. It is thus important to estimate the reasonable size for such seeds. The distribution of the Fig 13B helps to bound this size. Indeed, at a confidence level of 95% (resp. 99%), a seed of size at least $k = 108$ (resp. 76) exists in each read for low-GC species, and of size $k = 130$ (resp. 102) for high-GC species.

## Sequence-specific errors

Finally, we looked for *harmful k-mers*, words of size $k = 5$ that are frequently associated with a given type of error. We have produced logo and automata for each type of errors, considering the 5-mers before or after the error position. Fig 14 provides one of the most relevant results,

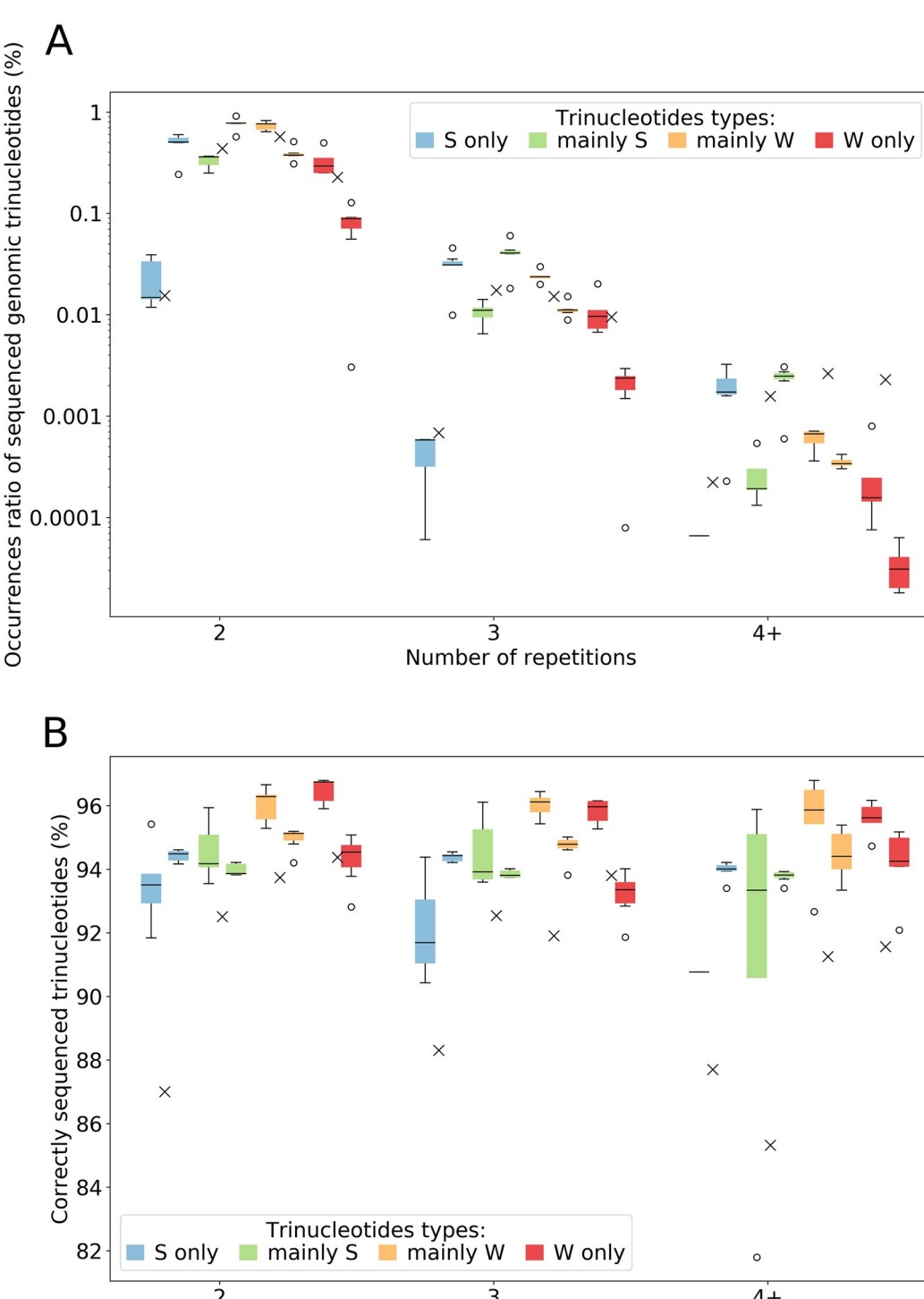

**Fig 12. Sequenced genomic trinucleotide repeats' occurrences and accuracy.** Trinucleotides with at least 4 repetitions are gathered within "4+" label. Global results are presented with boxplots for bacterial species (left boxplot for low-GC bacteria, and right one for high-GC) and crosses for human data (representing the mean of both datasets). For each bacterial species, only results for at least 100 occurrences are kept. Each color represents a trinucleotide category: blue for *S*-only, green for mostly-*S*, orange for mostly-*W*, and red for *W*-only trinucleotides. Results were computed on all reads for bacterial data, and on a subset of 100,000 reads for human data.

A

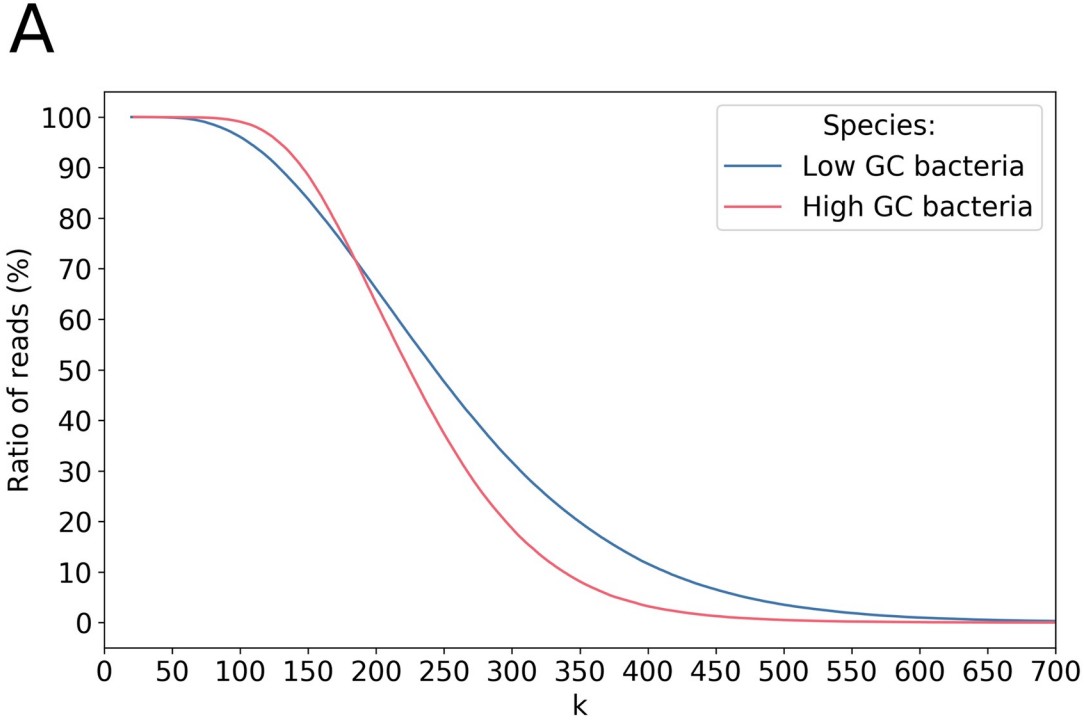

B

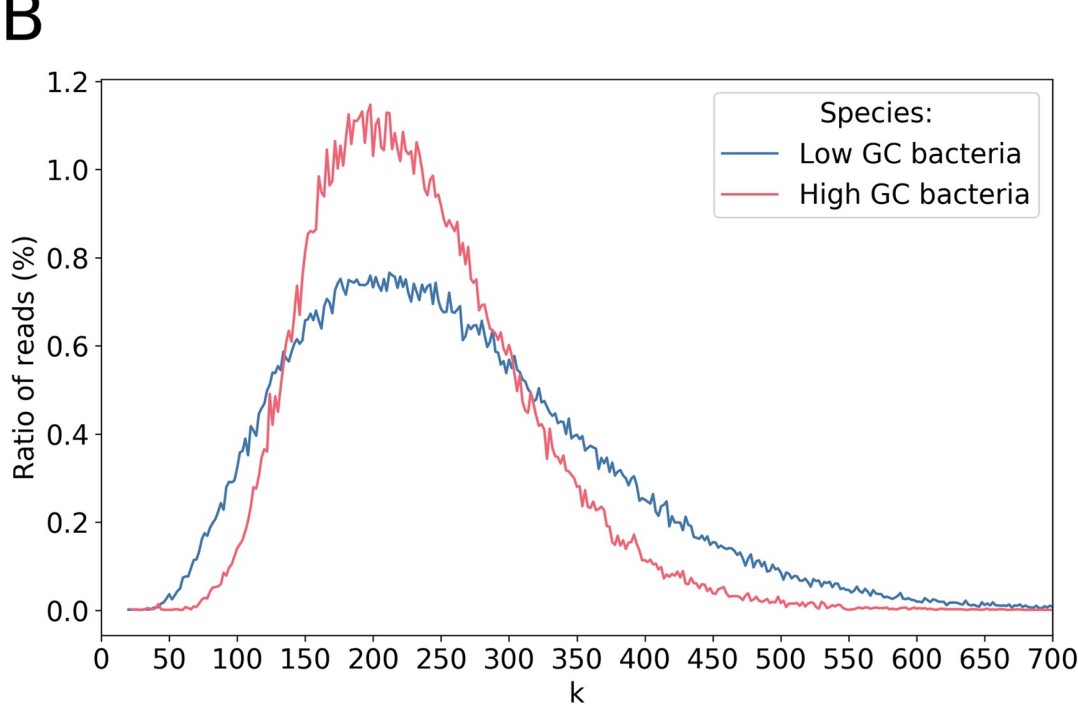

**Fig 13. Analysis of perfect k-mers for bacterial data.** Species are gathered based on their GC content.

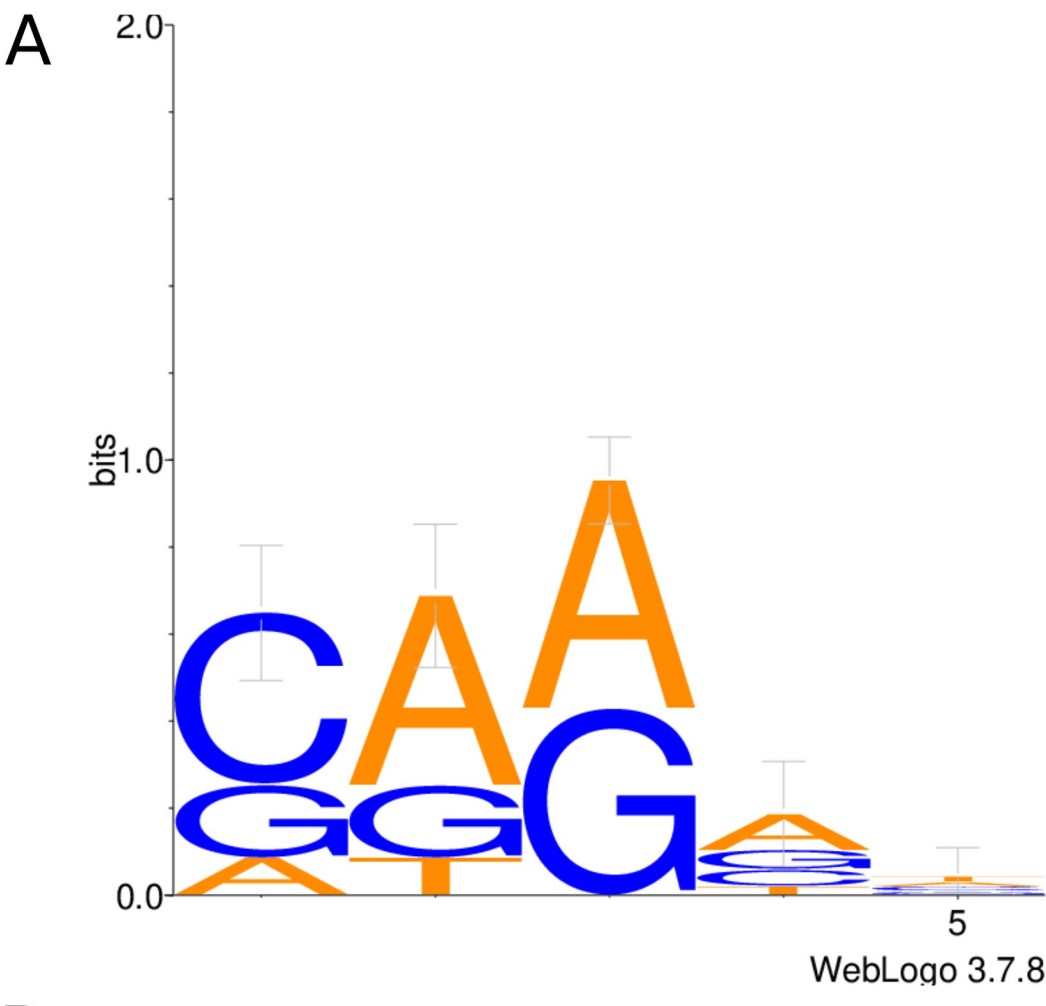

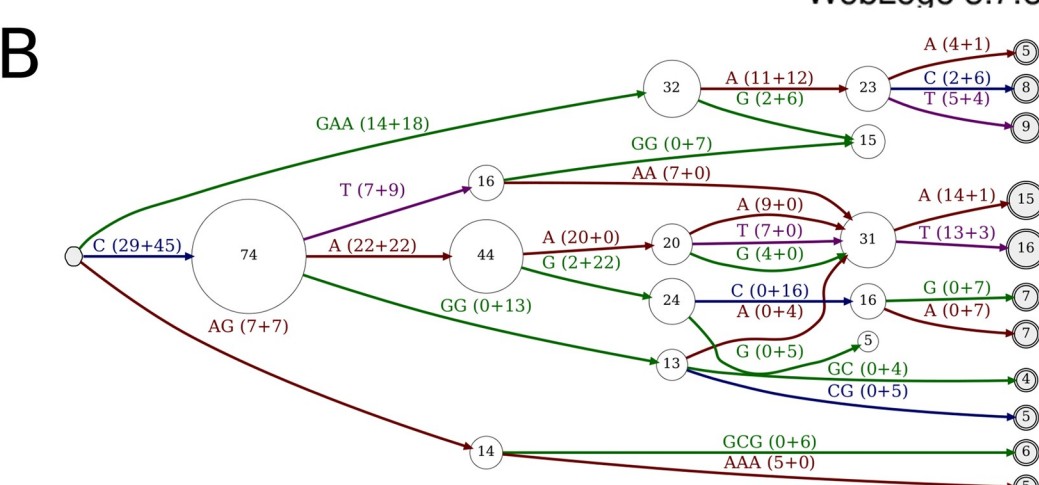

**Fig 14. Logo and finite automaton representing 5-mers before deletion errors in non-homopolymeric region, in bacterial species. A**: logo representation of 5-mers. **B**: automaton representation of 5-mers. Each node represents a position, and edges the letters in the k-mer. The initial node (on the very left side of the automaton, in grey) is the k-mer start. The final nodes (on the very right side of the automaton, in grey and double circled) are k-mer ends. For each edge, the weight (*i. e.* number of k-mers supporting the edge) is detailed as the sum for low- and high-GC species. Node size is proportional to the total weight of input edges, displayed inside nodes. Only nodes having a total weight of at least 4 are displayed.

concerning 5-mers after deletion errors (the main source of errors) in non-homopolymeric regions.

According to the logo representation, the most common pattern found right after deletions in non homopolymeric regions are k-mers starting with C or G, followed by two A and/or G. The automaton enables to precise these patterns: two of them is mainly found for high-GC bacteria (CGG and CAG), one is specific to low-GC bacteria (CAA), and one occurs for both categories (GAA). Note also that patterns like "GGG", fully compatible with the logo, do not appear in harmful kmers and are not present in the automaton.

## SeqFaiLR: A pipeline for analysis of sequencing accuracy evolution

In this work, we have covered a wide range of error profile features for the ONT MinION sequencer. Due to regular improvements, this snapshot is bound to evolve with future versions in chemistry and basecallers. Thus, we created a pipeline called SeqFaiLR (**Seq**uencing **Fai**lures of **L**ong **R**ead data) in order to enable the monitoring of these future developments. This pipeline computes the main analysis and figures of this article, for a given set of fastq files and references to be aligned with. As the required input is a set of fastq files, this pipeline could be used for various basecallers and any sequencing technology, although some analysis (such as repeated regions' sequencing accuracy) may not be relevant for non long read sequencing data. SeqFaiLR is available on github https://github.com/cdelahaye/SeqFaiLR.

## Conclusion

Due to the amount of data generated, fast5 files describing the original signal are rarely available for nanopore sequencing. For this reason, we focused mainly in this study on fastq files from two basecallers for which a majority of data are currently available, completing some of the findings with an analysis of the electrical signal. It concerns Guppy version 3.3.3 and 4.2.2, the most recent one, together with the best chemistry at the time of this study, R9.4.1. Since the chemistry changes much slower than the software part, we can only advise to keep the raw signal data because even years later, the evolution of the basecallers allows to significantly improve the accuracy of the read sequences. A comparison between the HAC and FAST basecalling modes of Guppy showed that the former produces more accurate reads, and we also clearly recommend using the HAC version if possible. The gain is modest at a global level, but accuracy increases significantly for homopolymer sequencing. Recently, ONT announced a soon to come release of a new basecaller called "Bonito", which will enable users to train the basecaller on their own datasets, thereby increasing the sequencing accuracy even further. At the time of writing, Bonito is only released as a beta version. This research has revealed several important features of MinION sequencer's sequencing errors. It is all the more necessary since the technology provider, Oxford Technology Nanopore, communicates little about the precise characteristics of its devices and softwares and does not offer the software it distributes in open source. It includes results concerning RNA direct sequencing.

We have first established that the quality score is strongly correlated to the error rate within reads. This can be very useful for tasks such as read filtering or *de novo* assembly, where a confidence score could be associated with overlap detection on this basis. The quality score is also linked to the amount of reads in a quasi-linear way for practical quality scores (*e.g.* in a 10–20 range) and a trade-off may be found depending on the initial number of reads and the desired coverage.

The second important point is that, even with PCR-free experiments, ONT sequencing is very sensitive to the GC content of reads. High-GC content reads have lower accuracy. This effect is accompanied by another bias that tends to make substitution errors towards A and T

nucleotides, thus decreasing the GC content of reads. For methylated bases, the error profile is also different depending on whether one considers low- or high-GC bacteria.

A third finding is that sequencing short repeated regions such as homopolymers as well as more complex STR is still challenging. About half of sequencing errors are due to homopolymers. Generally speaking, homopolymers and STR length tend to be underestimated, resulting in many deletion errors.

Another result is that analysis of perfect k-mers indicates that most reads contain perfect k-mers of size at least 100 bases, which could be helpful to assess which size of k-mers can be used for assembly. A last point concerns the behavior of the sequencer with respect to particular subsequences called SSE (sequence specific errors). Harmful k-mers after deletions in non-homopolymeric regions tend to be highly related to the GC-content of the species, and to be mainly made of C and G bases.

We hope the results presented in this study will provide guidance for better tuning of new basecallers or read simulators and future improvements in correction tools. As a by-product of this study we released a package integrating computations performed for this paper, which extends the scope of NanoOK [12] by providing a more complete picture of sequencing errors. This analysis can therefore be reused on new data and can be adapted to the evolution of the technology. It will provide a better understanding of the behaviour of future nanopore sequencers, basecallers and chemicals that will emerge.

## Supporting information

**S1 Fig. Quality scores along sequenced reads, for each dataset.** Top panel shows mean quality scores for relative position in read (gathered by %). Bottom panels zoom on both ends of reads, for the *n* first and last bases.
(TIF)

**S2 Fig. Ratio of well-sequenced homopolymers depending on their reference length and on basecaller mode, for bacterial data.** Results are split according to basecalling mode (HAC or FAST) and bacterial GC content (low or high).
(TIF)

**S3 Fig. Quality score as a function of GC content on a sliding 100-base window along reads.** Note the drop of about 1.5 in quality around the central GC value. The outlier species is *K. pneumoniae* INF032.
(TIF)

**S4 Fig. Mean error rate of reads depending on their quality score, for bacterial and human data.** Quality scores are rounded to the first decimal value. The dotted black line represents the expected Phred score relationship between quality score and error rate, other lines represent results obtained for our studied species. Results were computed all bacterial aligned reads, and on 100,000 aligned reads for each human dataset. Only values supported for at last *n* reads are shown (*n* = 10 for bacterial data, *n* = 10, 000 for human data).
(TIF)

**S5 Fig. Distribution of number of reads according to their error rate, for a range of quality values.** Results are computed on bacterial datasets.
(TIF)

**S6 Fig. Error rate and read number losses between basecaller quality thresholds 7 and 10.** Read number loss is computed as the difference in number of reads between thresholds 7 and 10, divided by the number of reads for threshold 7. Error rate loss is computed as a simple

difference in error rates between the two thresholds. Results are for bacterial datasets (colored dots). The black solid line shows linear regression.
(TIF)

**S7 Fig. Ratio of well-sequenced homopolymers as a function of their reference length, detailed for each of the four bases, for high-GC bacteria.** The base of the homopolymer does not strongly influence its sequencing accuracy, for length 2. However, for higher length, A- and T- based homopolymers are better sequenced than C- and G- ones. This trend is similar for low-GC bacteria and for human datasets.
(TIF)

**S8 Fig. Distribution of heteropolymer lengths in all reference genomes (forward strand only).** Symmetric dinucleotides (*e.g.* AC and CA) have been pooled. The scale is semi-logarithmic.
(TIF)

**S9 Fig. Errors in sequenced heteropolymer lengths.** Species grouped in 3 categories: low, high GC content, and human. Scale is semi-logarithmic. Dotted line represents expected length.
(TIF)

**S10 Fig. Error rates depending on translocation speed.** Computed on sliding windows of length 25 bases.
(TIF)

**S11 Fig. A- Tombo visualisation of 4-bases A homopolymer (centered)**. The homopolymer is rather short, and the associated signal is quite well segmented, which enables to easily delineate each base of the homopolymer. **B- Tombo visualisation of 6-bases A homopolymer (centered)**. The homopolymer is longer than in **A-**, and the associated signal is harder to segment, which complicates the separation between each base. **C- Tombo visualisation of 4-bases A homopolymer**. The homopolymer is surrounded by G bases, for which the signal value is closer than for **A-** where the homopolymer was surrounded by T bases. This imply less variation in signal values, thus resulting in a more blurred signal.
(TIF)

## Acknowledgments

All data were processed on the GenOuest bioinformatics platform (Biogenouest): https://www.genouest.org/. We thank E. Roux for having produced experimental data on *S. thermophilus*. We also thank J.M. Aury for providing us RNA dataset.

## Author Contributions

**Conceptualization:** Clara Delahaye, Jacques Nicolas.

**Formal analysis:** Clara Delahaye.

**Investigation:** Clara Delahaye, Jacques Nicolas.

**Methodology:** Clara Delahaye, Jacques Nicolas.

**Project administration:** Jacques Nicolas.

**Software:** Clara Delahaye.

**Supervision:** Jacques Nicolas.

**Validation:** Clara Delahaye, Jacques Nicolas.

**Visualization:** Clara Delahaye, Jacques Nicolas.

**Writing – original draft:** Clara Delahaye, Jacques Nicolas.

**Writing – review & editing:** Clara Delahaye, Jacques Nicolas.

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
