## [Decision Letter · Decision Letter 0]

19 May 2021

PONE-D-21-06109

Sequencing DNA with nanopores: troubles and biases.

PLOS ONE

Dear Dr. Delahaye,

Thank you for submitting your manuscript to PLOS ONE. After careful consideration, we feel that it has merit but does not fully meet PLOS ONE’s publication criteria as it currently stands. Therefore, we invite you to submit a revised version of the manuscript that addresses the points raised during the review process.

We look forward to receiving your revised manuscript.

Kind regards,

Eduardo Andrés-León

Academic Editor

PLOS ONE

Journal Requirements:

Comments are attached and included here:

Synopsis:

The goal of this paper is to identify the error landscape of ONT’s Nanopore platform. The authors identify deletions as the primary error type, and find an association between GC content and error rate. Mismatches are also identified as a significant source of error and are found to be potentially associated with base modifications. Nanopore QC is also empirically calculated and found to be an overestimation relative to the anticipated error rate. The authors also perform an analysis to show that harmful kmers exist that contribute to these error rates. The error-analysis pipeline used in this work is also provided.

Overview:

The paper seeks to analyze the error landscape of Nanopore sequencing and draws on a wealth of concepts developed in the analysis of the error landscape of second-generation sequencing. Although some error features of Nanopore and second-generation sequencing overlap, such that both technologies struggle with homopolymers and repeated kmers, other features are unique to Nanopore sequencing. Two key properties characterize Nanopore sequencing that distinguish it from second-generation Illumina sequencing: the changing voltages induced by the translocation of the nucleic acid molecule through the pore, and the very long reads that are produced. Unfortunately, with the exception of the base modification analysis, this paper focuses on neither of these aspects, instead focusing on error landscapes mostly derived from the history of error detection using second-generation sequencing.

The authors’ main measure of error is Nanopore QC, despite this quantification being intentionally obscured by ONT and, as the authors point out, inflated relative to the anticipated Phred score. An analysis of the raw signal (“Fast5 analysis” as described below) would help shed light on the mechanisms that drive these error-rates and determine whether homopolymers or harmful kmers alter translocation rates leading to the observed errors.

While the authors do describe an analysis to identify ‘local effects’ this is actually a comparison between genomes and not a local sequence analysis. There are several pieces of evidence that suggest local effects could be playing a role (“soft clipped”, “possible local effects”, and “inverted repeats” as described below). An analysis of local effects may also help identify the source of sequencing errors associated with harmful kmers.

Major Issues:

“Soft clipped” (Pg 4, lines 134, 135)

Here the authors state that reads that are over 50% soft-clipped are removed. However, if I understand the protocol correctly, reads have already been filtered if their Nanopore QC score is less than 10 (line 124). This suggests that there are reads with 50% soft-clipping that have QC scores 10 and up. This begs several questions that are central to the authors stated intent of understanding ONT error rates. Namely:

How many reads were removed?

If the read itself has QC >= 10, do the soft clipped bases also have high QC?

If the soft clipped bases are the same between reads? (ie. assuming that at least two reads overlap - do the soft clipped bases align to each other (suggesting a true variation from the reference genome) or do they disagree (suggesting the sequencer is failing to accurately sequence anything, ie “noise”)?

If the soft-clipped reads are noise it would be very important to confirm that they have low QC - especially given the repeated failure of ONT to explain what Nanopore QC actually means.

“Possible local effects” (Pg 7, lines 233 - 253, Figure 1)

The authors seek to determine if there are local effects that can generate changes in error frequency. To accomplish this they scan the linearized bacterial genomes and plot the results and come to the conclusion that there may be no such effecst. However, there are several issues with this interpretation.

First, the bacterial genomes are set to align at the beginning of the FASTA file - but that is an arbitrary position for circular bacterial genomes. While bacterial genomes do have a reference point that could be used for alignment (ie. replication origin (oriC)) that is not used here and the results are impossible to compare across species as a result. It would be much better to center the genomes at the oriC and then compare.

Second, actual local effects may also be specific to a genome as not all the species have the same gene content and synteny. There are notable peaks amongst the noise -”with some possible local peaks” (line 246) - but to speak to the question of possible local effects these peaks would need to be interrogated (do they coincide with the oriC, rRNA, low-complexity regions?) and then compared across species.

“Inverted repeats” (Pg. 15, lines 551-555)

If I understand the work being referenced, these were very long inverted repeats produced from a CNV mutation. The absence of detection from these genomes is not indicative of a corrected basecalling method as these CNVs may not be present within these genomes. Unless the authors observed inverted repeat CNVs in these bacteria using the latest version of the basecaller, and observe CNV sequencing failures using a previous version of the basecaller it seems unwarranted to state that a solution has been produced.

“Fast5 analysis” (Pg. 16, lines 582, 583)

The authors state that because few Fast5 files are publicly available they are focusing on the post base-calling FASTQ files. While I am sympathetic to the lack of Fast5 file support in the public repositories - it appears that the Fast5 files are available for the data sets the authors have chosen to analyze. As several researchers have already shown (and as stated by the authors) analysis of the Fast5 files allows for the identification of changes in translocation rate. Since this would generate deletion errors (which are the primary contributor to error rates) it seems appropriate to investigate these in this case. Furthermore, an analysis of the Fast5 files would enable the authors to identify correlations in changes in error rate with the sequencing process itself.

Minor Issues:

Abstract - “Oxford Nanopore Technologies” should be used once before being abbreviated to “ONT”.

Line 5 - Replace “proposes” with “that uses”

Line 140 - Very interesting insight. Did this change the observed error rates?

Line 160 - Replace “couple” with “pair”

Line 258 - Replace “in average” with “on average”

Line 278 - Replace “errors” with “error”

Line 290-292 - Unclear if the observations did change when you used reverse sequence or they did not change. (See comment for Line 140, above).

Line 494 - “CG heteropolymers for which the error rate is much higher.” This is a very interesting result and yet it seems to be unaddressed. Is the higher error rate of CG in human data indicative of the CpG rates?

Line 551 - Not clear why this paragraph is under the ‘Perfect Kmer section’, potentially a formatting issue?

Figure 7 - From this figure it is difficult to identify if the homopolymers are causing the observed abundances in sequencing errors or if it is the frequency of these homopolymers in the genome. By which I mean, are A/T homopolymers more error prone or just more common? Does normalizing by frequency in the genome (or frequency of reads, as in Figure 8) remove the apparent difference between A/T and C/G?

Figure 9 - The plot is difficult to understand, is the Y-axis incorrectly labelled?

Reviewers' comments:

Reviewer's Responses to Questions

**Comments to the Author**

1. Is the manuscript technically sound, and do the data support the conclusions?

Reviewer #1: Partly

2. Has the statistical analysis been performed appropriately and rigorously? 

Reviewer #1: Yes

3. Have the authors made all data underlying the findings in their manuscript fully available?

Reviewer #1: Yes

4. Is the manuscript presented in an intelligible fashion and written in standard English?

Reviewer #1: Yes

5. Review Comments to the Author

Reviewer #1: Review Comments to the Author

Please use the space provided to explain your answers to the questions above. You may also include additional comments for the author, including concerns about dual publication, research ethics, or publication ethics. (Please upload your review as an attachment if it exceeds 20,000 characters) (Limit 200 to 20000 Characters)

Please see my attached review.

6. PLOS authors have the option to publish the peer review history of their article (what does this mean?). If published, this will include your full peer review and any attached files.

Reviewer #1: No

---

## [Author Response · Author response to Decision Letter 0]

12 Jul 2021

The response to reviewer and editor has been uploaded in the file named "Rebuttal_letter.pdf".

---

## [Decision Letter · Decision Letter 1]

6 Sep 2021

Sequencing DNA with nanopores: troubles and biases.

PONE-D-21-06109R1

Dear Dr. Delahaye,

We’re pleased to inform you that your manuscript has been judged scientifically suitable for publication and will be formally accepted for publication once it meets all outstanding technical requirements.

Kind regards,

Eduardo Andrés-León

Academic Editor

PLOS ONE

Additional Editor Comments (optional):

Reviewers' comments:

Reviewer's Responses to Questions

**Comments to the Author**

1. If the authors have adequately addressed your comments raised in a previous round of review and you feel that this manuscript is now acceptable for publication, you may indicate that here to bypass the “Comments to the Author” section, enter your conflict of interest statement in the “Confidential to Editor” section, and submit your "Accept" recommendation.

Reviewer #2: (No Response)

2. Is the manuscript technically sound, and do the data support the conclusions?

Reviewer #2: Yes

3. Has the statistical analysis been performed appropriately and rigorously? 

Reviewer #2: Yes

4. Have the authors made all data underlying the findings in their manuscript fully available?

Reviewer #2: Yes

5. Is the manuscript presented in an intelligible fashion and written in standard English?

Reviewer #2: Yes

6. Review Comments to the Author

Reviewer #2: This reviewer thinks that this manuscript “Sequencing DNA with nanopores: troubles and biases.” is acceptable for publication. The manuscript studied Nanopore sequencing error biases on both bacterial and human DNA reads. The manuscript Is technically sound, and their data analysis support the conclusions. This reviewer found the manuscript is interesting and the topic is very important for the development and improvement of future sequencing devices.

7. PLOS authors have the option to publish the peer review history of their article (what does this mean?). If published, this will include your full peer review and any attached files.

Reviewer #2: No

---

## [Editor Report · Acceptance letter]

22 Sep 2021

PONE-D-21-06109R1 

Sequencing DNA with nanopores: troubles and biases. 

Dear Dr. Delahaye:

I'm pleased to inform you that your manuscript has been deemed suitable for publication in PLOS ONE. Congratulations! Your manuscript is now with our production department. 

Kind regards, 

on behalf of

Dr. Eduardo Andrés-León 

Academic Editor

PLOS ONE